# PARTIAL INFORMATION AS FULL:
# REWARD IMPUTATION WITH SKETCHING IN BANDITS

## ABSTRACT

We focus on the setting of contextual batched bandit (CBB), where a batch of rewards is observed from the environment in each episode. But these rewards are partial-information feedbacks where the rewards of the non-executed actions are unobserved. Existing approaches for CBB usually ignore the potential rewards of the non-executed actions, resulting in feedback information being underutilized. In this paper, we propose an efficient reward imputation approach using sketching in CBB, which completes the unobserved rewards with the imputed rewards approximating the full-information feedbacks. Specifically, we formulate the reward imputation as a problem of imputation regularized ridge regression, which captures the feedback mechanisms of both the non-executed and executed actions. To reduce the time complexity of reward imputation on a large batch of data, we use randomized sketching for solving the regression problem of imputation. We prove that the proposed reward imputation approach obtains a relative-error bound for sketching approximation, achieves an instantaneous regret with a controllable bias and a smaller variance than that without reward imputation, and enjoys a sublinear regret bound against the optimal policy. Moreover, we present two extensions of our approach, including the rate-scheduled version and the version for nonlinear rewards, which makes our approach more feasible. Experimental results demonstrated that our approach can outperform the state-of-the-art baselines on a synthetic dataset, the Criteo dataset, and a dataset from a commercial app.

## 1 INTRODUCTION

Contextual bandits have been widely used in real-world sequential decision-making problems (Li et al., 2010; Lan & Baraniuk, 2016; Yom-Tov et al., 2017; Yang et al., 2021), where the agent updates the decision-making policy fully online (i.e., at each step) according to the context and corresponding reward feedback so as to maximize the cumulative reward. In this paper, we consider a more complex setting—*contextual batched bandits* (CBB), where the decision process is partitioned into $N$ episodes, the agent interacts with the environment for $B$ steps in one episode, collects the reward feedbacks and contexts at the end of the episode, and then updates the policy using the collected data for the next episode. CBB is more practical in some real-world applications, since updating the policy once receiving the reward feedback is rather unrealistic due to its high computational cost and decision instability.

In bandit settings, it is inevitable that the environment only reveals the rewards of the executed actions to the agent as the feedbacks, while hiding the rewards of non-executed actions. We refer to this category of limited feedback as the *partial-information feedback* (also called "bandit feedback"). Existing batched bandit approaches in the CBB setting discard the information contained in the potential rewards of the non-executed actions, address the problem of partial-information feedback using an exploitation-exploration tradeoff on the context space and reward space (Han et al., 2020; Zhang et al., 2020).But in the CBB setting, the agent usually estimates and maintains reward models for the action-selection policy, and the potential rewards of the non-executed actions have been somehow captured by the policy. This additional reward structure information is estimated and available in each episode, however, are not utilized by existing batched bandit approaches.

In contextual bandit settings where the policy is updated fully online, several bias-correction approaches have been introduced to address the partial-information feedback. Dimakopoulou et al.

(2019) presented linear contextual bandits integrating the balancing approach from causal inference, which reweight the contexts and rewards by the inverse propensity scores. Chou et al. (2015) designed pseudo-reward algorithms for contextual bandits, which use a direct method to estimate the unobserved rewards for the upper confidence bound (UCB) strategy. Kim & Paik (2019) focused on the feedback bias-correction for LASSO bandit with high-dimensional contexts, and applied the doubly-robust approach to the reward modification using average contexts. Although these approaches have been demonstrated to be effective in contextual bandit settings, little efforts have been spent to address the under-utilization of partial-information feedback in the CBB setting.

Theoretical and experimental analyses in Section 2 indicate that better performance of CBB is achievable if the rewards of the non-executed actions can be received. Motivated by these observations, we propose a novel reward imputation approach for the non-executed actions, which mimics the reward generation mechanisms of environments. We conclude our contributions as follows.

• To fully utilize feedback information in CBB, we formulate the reward imputation as a problem of imputation regularized ridge regression, where the policy can be updated efficiently using sketching.

• We prove that our reward imputation approach obtains a relative-error bound for sketching approximation, achieves an instantaneous regret with a controllable bias and a smaller variance than that without reward imputation, has a lower bound of the sketch size independently of the overall number of steps, enjoys a sublinear regret bound against the optimal policy, and reduces the time complexity from $O(Bd^2)$ to $O(cd^2)$ for each action in one episode, where $B$ denotes the batch size, $c$ the sketch size, and $d$ the *dimensionality of inputs*, satisfying $d < c < B$.

• We present two practical variants of our reward imputation approach, including the rate-scheduled version in which the imputation rate is set without tuning, and the version for nonlinear rewards.

• We carried out extensive experiments on the synthetic data, public benchmark, and the data collected from a real commercial product to demonstrate our performance, empirically analyzed the influence of different parameters, and verified the correctness of the theoretical results.

**Related Work.** Recently, batched bandit has become an active research topic in statistics and learning theory including 2-armed bandit (Perchet et al., 2016), multi-armed bandit (Gao et al., 2019; Zhang et al., 2020; Wang & Cheng, 2020), and contextual bandit (Han et al., 2020; Ren & Zhou, 2020; Gu et al., 2021). Han et al. (2020) defined linear contextual bandits, and designed UCB-type algorithms for both stochastic and adversarial contexts, where true rewards of different actions have the same parameters. Zhang et al. (2020) provided methods for inference on data collected in batches using bandits, and introduced a batched least squares estimator for both multi-arm and contextual bandits. Recently, Esfandiari et al. (2021) proved refined regret upper bounds of batched bandits in stochastic and adversarial settings. There are several recent works that consider similar settings to CBB, e.g., episodic Markov decision process (Jin et al., 2018), LASSO bandits (Wang & Cheng, 2020). Sketching is another related technology that compresses a large matrix to a much smaller one by multiplying a (usually) random matrix with certain properties (Woodruff, 2014), which has been used in online convex optimization (Calandriello et al., 2017; Zhang & Liao, 2019).

## 2 PROBLEM FORMULATION AND ANALYSIS

First, we introduce some notations. Let $[x] = \{1, 2, \ldots, x\}$, $\mathcal{S} \subseteq \mathbb{R}^d$ be the context space, $\mathcal{A} = \{A_j\}_{j \in [M]}$ the action space containing $M$ actions, $[\boldsymbol{A}; \boldsymbol{B}] = [\boldsymbol{A}^\mathsf{T}, \boldsymbol{B}^\mathsf{T}]^\mathsf{T}$, $\|\boldsymbol{A}\|_\mathrm{F}$, $\|\boldsymbol{A}\|_1$ $\|\boldsymbol{A}\|_2$ denote the Frobenius norm, 1-norm, and spectral norm of a matrix $\boldsymbol{A}$, respectively, $\|\boldsymbol{a}\|_1$ and $\|\boldsymbol{a}\|_2$ be the $\ell_1$-norm and the $\ell_2$-norm of a vector $\boldsymbol{a}$, $\sigma_{\min}(\boldsymbol{A})$ and $\sigma_{\max}(\boldsymbol{A})$ denote the minimum and maximum of the of singular values of $\boldsymbol{A}$. In this paper, we focus on the setting of *Contextual Batched Bandits* (CBB) (see Algorithm 1), where the decision process is partitioned into $N$ episodes, and in each episode, CBB consists of two phases: 1) the *policy updating* approximates the optimal policy based on the received contexts and rewards; 2) the *online decision* selects actions for execution following the updated and fixed policy $p$ for $B$ steps (also called the *batch size* is $B$), and stores the context-action pairs and the observed rewards of the executed actions into a data buffer $\mathcal{D}$. The reward $R$ in CBB is a *partial-information feedback* where rewards are unobserved for the non-executed actions.

Different from existing batch bandit setting (Han et al., 2020; Esfandiari et al., 2021), where the true reward feedbacks for all actions are controlled by the same parameter vector while the contexts received differ by actions at each step, we assume that in the CBB setting, the mechanism of true reward feedback differs by actions and the received context is shared by actions. Formally, for any context $\boldsymbol{s}_i \in \mathcal{S} \subseteq \mathbb{R}^d$ and action $A \in \mathcal{A}$, we assume that the expectation of the true reward $R_{i,A}^{\mathrm{true}}$

---

**Algorithm 1** Contextual Batched Bandit (CBB)

---

**INPUT:** Batch size $B$, number of episodes $N$, action space $\mathcal{A} = \{A_j\}_{j \in [M]}$, context space $\mathcal{S} \subseteq \mathbb{R}^d$
1: Initialize policy $p_0 \leftarrow \mathbf{1}/M$, sample data buffer $\mathcal{D}_1 = \{(\boldsymbol{s}_{0,b}, A_{I_{0,b}}, R_{0,b})\}_{b \in [B]}$ using initial policy $p_0$
2: **for** $n = 1$ **to** $N$ **do**
3:     Update the policy $p_n$ on $\mathcal{D}_n$  {Policy Updating}
4:     **for** $b = 1$ **to** $B$ **do**
5:         Observe context $\boldsymbol{s}_{n,b}$
6:         Choose $A_{I_{n,b}} \in \mathcal{A}$ following the updated policy $p_n(\boldsymbol{s}_{n,b})$  {Online Decision}
7:     **end for**
8:     $\mathcal{D}_{n+1} \leftarrow \{(\boldsymbol{s}_{n,b}, A_{I_{n,b}}, R_{n,b})\}_{b \in [B]}$, where $R_{n,b}$ denotes the reward of action $A_{I_{n,b}}$ on context $\boldsymbol{s}_{n,b}$
9: **end for**

---

is determined by an unknown action-specific *reward parameter vector* $\boldsymbol{\theta}_A^* \in \mathbb{R}^d$: $\mathbb{E}\left[R_{i,A}^{\text{true}} \mid \boldsymbol{s}_i\right] = \langle \boldsymbol{\theta}_A^*, \boldsymbol{s}_i \rangle$ (the linear reward will be extended to the nonlinear case in Section 5). This setting for reward feedback matches many real-world applications, e.g., each action corresponds to a different category of candidate coupons in coupon recommendation, and the reward feedback mechanism of each category differs due to the different discount pricing strategies.

Next, we provide deeper understandings of the influence of unobserved feedbacks on the performance of policy updating in the CBB setting. We first conducted an empirical comparison by applying the batch UCB policy (Han et al., 2020) to environments under different proportions of received reward feedbacks. In particular, the agent under full-information feedback can receive all the rewards of the executed and non-executed actions, called *Full-Information CBB* (FI-CBB) setting. From Figure 1, we can observe that the partial-information feedbacks could be damaging in terms of hurting the policy updating, and batched bandit could benefit from more reward feedbacks, where the performance of $80\%$ feedback is very close to that of FI-CBB.

Then, we demonstrate the difference of instantaneous regrets between the CBB and FI-CBB settings as in Theorem 1. The detailed description and proof of Theorem 1 can be found in Appendix A.

**Theorem 1.** *For any action $A \in \mathcal{A}$ and context $\boldsymbol{s}_i \in \mathcal{S}$, let $\boldsymbol{\theta}_A^n$ be the reward parameter vector estimated by the batched UCB policy in the $n$-th episode. The upper bound of instantaneous regret (defined by $|\langle \boldsymbol{\theta}_A^n, \boldsymbol{s}_i \rangle - \langle \boldsymbol{\theta}_A^*, \boldsymbol{s}_i \rangle|$) in the FI-CBB setting is tighter than that in CBB setting (i.e., using the partial-information feedback).*

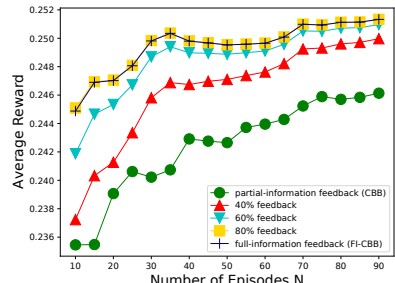

Figure 1: Average rewards of batch UCB policy (Han et al., 2020) under different proportions of received reward feedbacks, interacting with the synthetic environment in Section 6, where $x\%$ feedback means that $x\%$ of actions can receive their true rewards

From Theorem 1, we can conclude that the price paid for having access only to the partial-information feedbacks is the deterioration in the regret bound. Ideally, the policy would be updated using the full-information feedback. In CBB, however, the full-information feedback is unaccessible. Fortunately, in CBB, different reward parameter vectors need to be maintained and estimated separately for each action, and the potential reward structures of the non-executed actions have been somehow captured. So why don't we use these maintained reward parameters to estimate the unknown rewards for the non-executed actions? Next, we propose an efficient reward imputation approach that uses this additional reward structure information for improving the performance of the bandit policy.

## 3 EFFICIENT REWARD IMPUTATION FOR POLICY UPDATING

In this section, we present an efficient reward imputation approach tailored for policy updating in the CBB setting.

**Formulation of Reward Imputation.** Since the true reward parameters differ by actions in the CBB setting, we need to maintain a different estimated reward parameter vector for each action. As shown in Figure 2, in contrast to CBB that ignores the contexts and rewards of the non-executed steps of $A_j$, our reward imputation approach completes the missing values using the imputed contexts and rewards, approximating the full-information CBB setting. Specifically, in the $(n+1)$-th episode, for

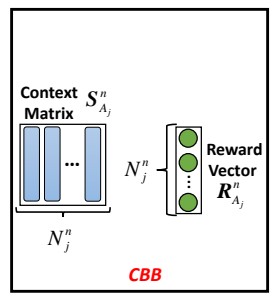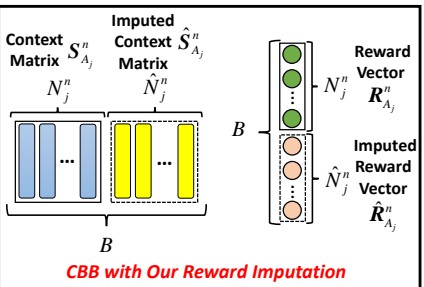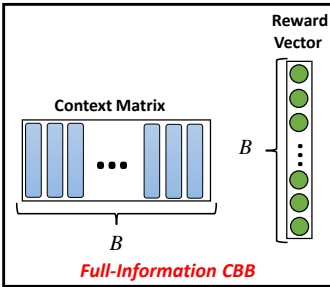

Figure 2: Comparison of the stored data corresponding to the action $A_j \in \mathcal{A}$ in CBB, CBB with our reward imputation, and full-information CBB, in the $(n+1)$-th episode

each action $A_j \in \mathcal{A}, j \in [M]$, we store context vectors and rewards corresponding to the steps in which the action $A_j$ is executed, into a *context matrix* $\boldsymbol{S}_{A_j}^n \in \mathbb{R}^{N_j^n \times d}$ and a *reward vector* $\boldsymbol{R}_{A_j}^n \in \mathbb{R}^{N_j^n}$, respectively, where $N_j^n$ denotes the number of steps (in episode $n+1$) in which the action $A_j$ is executed. Additionally, for any $A_j \in \mathcal{A}, j \in [M]$, we store context vectors corresponding to the non-executed steps of action $A_j$ (denote the number of non-executed steps by $\widehat{N}_j^n$, i.e., $\widehat{N}_j^n = B - N_j^n$), into an *imputed context matrix* $\widehat{\boldsymbol{S}}_{A_j}^n \in \mathbb{R}^{\widehat{N}_j^n \times d}$, and compute the *imputed reward vector* as follows:

$$\widehat{\boldsymbol{R}}_{A_j}^n = \{r_{n,1}(A_j), r_{n,2}(A_j), \ldots, r_{n,\widehat{N}_j^n}(A_j)\} \in \mathbb{R}^{\widehat{N}_j^n}, \quad j \in [M],$$

where $r_{n,b}(A_j) := \langle \bar{\boldsymbol{\theta}}_{A_j}^n, \boldsymbol{s}_{n,b} \rangle$ denotes the *imputed reward* for each step $b \in [\widehat{N}_j^n]$, and $\boldsymbol{s}_{n,b}$ is the $b$-th row of $\widehat{\boldsymbol{S}}_{A_j}^n$. Then, we obtain several block matrices by concatenating the context and reward matrices from the previous episodes: $\boldsymbol{L}_{A_j}^n = [\boldsymbol{S}_{A_j}^0; \cdots; \boldsymbol{S}_{A_j}^n] \in \mathbb{R}^{L_j^n \times d}$, $\boldsymbol{T}_{A_j}^n = [\boldsymbol{R}_{A_j}^0; \cdots; \boldsymbol{R}_{A_j}^n] \in \mathbb{R}^{L_j^n}$, $L_j^n = \sum_{k=0}^n N_j^k$, $\widehat{\boldsymbol{L}}_{A_j}^n = [\widehat{\boldsymbol{S}}_{A_j}^0; \cdots; \widehat{\boldsymbol{S}}_{A_j}^n] \in \mathbb{R}^{\widehat{L}_j^n \times d}$, $\widehat{\boldsymbol{T}}_{A_j}^n = [\widehat{\boldsymbol{R}}_{A_j}^0; \cdots; \widehat{\boldsymbol{R}}_{A_j}^n] \in \mathbb{R}^{\widehat{L}_j^n}$, $\widehat{L}_j^n = \sum_{k=0}^n \widehat{N}_j^k$. In the $(n+1)$-th episode, the *estimated parameter vector* $\bar{\boldsymbol{\theta}}_{A_j}^{n+1}$ of the imputed reward for action $A_j$ can be updated by solving the following *imputation regularized ridge regression*:

$$\bar{\boldsymbol{\theta}}_{A_j}^{n+1} = \underset{\boldsymbol{\theta} \in \mathbb{R}^d}{\arg\min} \underbrace{\left\| \boldsymbol{L}_{A_j}^n \boldsymbol{\theta} - \boldsymbol{T}_{A_j}^n \right\|_2^2}_{\text{Observed Term}} + \gamma \underbrace{\left\| \widehat{\boldsymbol{L}}_{A_j}^n \boldsymbol{\theta} - \widehat{\boldsymbol{T}}_{A_j}^n \right\|_2^2}_{\text{Imputation Term}} + \lambda \|\boldsymbol{\theta}\|_2^2, \quad n = 0, 1, \ldots, N-1, \quad (1)$$

where $\gamma \in [0, 1]$ is the *imputation rate* which controls the degree of reward imputation and measures a trade-off between bias and variance (Remark 1&2), $\lambda > 0$ is the regularization parameter, yielding

$$\bar{\boldsymbol{\theta}}_{A_j}^{n+1} = \left( \boldsymbol{\Psi}_{A_j}^{n+1} \right)^{-1} \left( \boldsymbol{b}_{A_j}^{n+1} + \gamma \hat{\boldsymbol{b}}_{A_j}^{n+1} \right), \quad (2)$$

that can be obtained by the closed least squares solution, where $\boldsymbol{\Psi}_{A_j}^{n+1} := \lambda \boldsymbol{I}_d + \boldsymbol{\Phi}_{A_j}^{n+1} + \gamma \widehat{\boldsymbol{\Phi}}_{A_j}^{n+1}$,

$$\boldsymbol{\Phi}_{A_j}^{n+1} = \boldsymbol{\Phi}_{A_j}^n + \boldsymbol{S}_{A_j}^{n\mathsf{T}} \boldsymbol{S}_{A_j}^n, \quad \boldsymbol{b}_{A_j}^{n+1} = \boldsymbol{b}_{A_j}^n + \boldsymbol{S}_{A_j}^{n\mathsf{T}} \boldsymbol{R}_{A_j}^n, \quad (3)$$

$$\widehat{\boldsymbol{\Phi}}_{A_j}^{n+1} = \eta \widehat{\boldsymbol{\Phi}}_{A_j}^n + \widehat{\boldsymbol{S}}_{A_j}^{n\mathsf{T}} \widehat{\boldsymbol{S}}_{A_j}^n, \quad \hat{\boldsymbol{b}}_{A_j}^{n+1} = \eta \hat{\boldsymbol{b}}_{A_j}^n + \widehat{\boldsymbol{S}}_{A_j}^{n\mathsf{T}} \widehat{\boldsymbol{R}}_{A_j}^n, \quad (4)$$

and $\eta \in (0, 1)$ is the *discount parameter* which controls how fast the previous imputed rewards are forgotten, and can help guaranteeing the regret bound in Theorem 2.

**Efficient Reward Imputation using Sketching.** As shown in the first 4 columns in Table 1, the overall time complexity of the imputation for each action is $O(Bd^2)$ in each episode, where $B$ represents the batch size, and $d$ the dimensionality of the input. Thus, for all the $M$ actions in one episode, reward imputation increases the time complexity from $O(Bd^2)$ of the approach without imputation to $O(MBd^2)$. To address this issue, we design an efficient reward imputation approach using sketching, which reduces the time complexity of each action in one episode from $O\left(Bd^2\right)$ to $O\left(cd^2\right)$, where $c$ denotes the *sketch size* satisfying $d < c < B$ and $cd > B$. Specifically, in the $(n+1)$-th episode, the imputation regularized ridge regression Eq.(1) can be approximated by a *sketched ridge regression* as

$$\tilde{\boldsymbol{\theta}}_{A_j}^{n+1} = \underset{\boldsymbol{\theta} \in \mathbb{R}^d}{\arg\min} \left\| \boldsymbol{\Pi}_{A_j}^n \left( \boldsymbol{L}_{A_j}^n \boldsymbol{\theta} - \boldsymbol{T}_{A_j}^n \right) \right\|_2^2 + \gamma \left\| \widehat{\boldsymbol{\Pi}}_{A_j}^n \left( \widehat{\boldsymbol{L}}_{A_j}^n \boldsymbol{\theta} - \widehat{\boldsymbol{T}}_{A_j}^n \right) \right\|_2^2 + \lambda \|\boldsymbol{\theta}\|_2^2, \quad (5)$$

Table 1: The time complexities of the original reward imputation in Eq.(1) (first 4 columns) and the reward imputation using sketching in Eq.(5) (last 4 columns) for action $A_j$ in the $(n+1)$-th episode, where $N_j^n$ $(\widehat{N}_j^n)$ denotes the number of steps in which the action $A_j$ is executed (non-executed) in episode $n+1$, $\widehat{N}_j^n + N_j^n = B$, and the sketch size $c$ satisfying $d < c < B$ and $cd > B$ (MM: matrix multiplication; MI: matrix inversion; Overall: overall time complexity for action $A_j$ in one episode)

| Original reward imputation in Eq.(1) | | | | Reward imputation using sketching in Eq.(5) | | | |
|---|---|---|---|---|---|---|---|
| Item | Operation | Eq. | Time | Item | Operation | Eq. | Time |
| $\boldsymbol{\Phi}_{A_j}^{n+1}, \widehat{\boldsymbol{\Phi}}_{A_j}^{n+1}$ | MM | (3), (4) | $O(Bd^2)$ | $\boldsymbol{G}_{A_j}^{n+1}, \widehat{\boldsymbol{G}}_{A_j}^{n+1}$ | MM | (7), (8) | $O(cd^2)$ |
| $\boldsymbol{b}_{A_j}^{n+1}, \hat{\boldsymbol{b}}_{A_j}^{n+1}$ | MM | (3), (4) | $O(Bd)$ | $\boldsymbol{p}_{A_j}^{n+1}, \hat{\boldsymbol{p}}_{A_j}^{n+1}$ | MM | (7), (8) | $O(cd)$ |
| $(\boldsymbol{\Psi}_{A_j}^{n+1})^{-1}$ | MI | (2) | $O(d^3)$ | $(\boldsymbol{W}_{A_j}^{n+1})^{-1}$ | MI | (6) | $O(d^3)$ |
| | – | | | $\boldsymbol{\Gamma}_{A_j}^n, \boldsymbol{\Lambda}_{A_j}^n$ | Sketching | – | $O(N_j^n d)$ |
| | – | | | $\widehat{\boldsymbol{\Gamma}}_{A_j}^n, \widehat{\boldsymbol{\Lambda}}_{A_j}^n$ | Sketching | – | $O(\widehat{N}_j^n d)$ |
| Overall | – | – | $O(Bd^2)$ | Overall | – | – | $O(cd^2)$ |

where $\tilde{\boldsymbol{\theta}}_A^{n+1}$ denotes the estimated parameter vector of the imputed reward using sketching for action $A \in \mathcal{A}$, $\boldsymbol{C}_{A_j}^n \in \mathbb{R}^{c \times N_j^n}$ and $\widehat{\boldsymbol{C}}_{A_j}^n \in \mathbb{R}^{c \times \widehat{N}_j^n}$ are the *sketch submatrices* for the observed term and the imputation term, respectively, and the *sketch matrices* for the two terms can be represented as

$$\boldsymbol{\Pi}_{A_j}^n = \left[ \boldsymbol{C}_{A_j}^0, \boldsymbol{C}_{A_j}^1, \cdots, \boldsymbol{C}_{A_j}^n \right] \in \mathbb{R}^{c \times L_j^n}, \quad \widehat{\boldsymbol{\Pi}}_{A_j}^n = \left[ \widehat{\boldsymbol{C}}_{A_j}^0, \widehat{\boldsymbol{C}}_{A_j}^1, \cdots, \widehat{\boldsymbol{C}}_{A_j}^n \right] \in \mathbb{R}^{c \times \widehat{L}_j^n}.$$

We denote the sketches of the context matrix and the reward vector by $\boldsymbol{\Gamma}_{A_j}^n := \boldsymbol{C}_{A_j}^n \boldsymbol{S}_{A_j}^n \in \mathbb{R}^{c \times d}$ and $\boldsymbol{\Lambda}_{A_j}^n := \boldsymbol{C}_{A_j}^n \boldsymbol{R}_{A_j}^n \in \mathbb{R}^c$, the sketches of the imputed context matrix and the imputed reward vector by $\widehat{\boldsymbol{\Gamma}}_{A_j}^n := \widehat{\boldsymbol{C}}_{A_j}^n \widehat{\boldsymbol{S}}_{A_j}^n \in \mathbb{R}^{c \times d}$ and $\widehat{\boldsymbol{\Lambda}}_{A_j}^n := \widehat{\boldsymbol{C}}_{A_j}^n \widehat{\boldsymbol{R}}_{A_j}^n \in \mathbb{R}^c$, and obtain the solution of Eq.(5):

$$\tilde{\boldsymbol{\theta}}_{A_j}^{n+1} = \left( \boldsymbol{W}_{A_j}^{n+1} \right)^{-1} \left( \boldsymbol{p}_{A_j}^{n+1} + \gamma \hat{\boldsymbol{p}}_{A_j}^{n+1} \right), \tag{6}$$

where $\eta \in (0, 1)$ denotes the discount parameter, $\boldsymbol{W}_{A_j}^{n+1} := \lambda \boldsymbol{I}_d + \boldsymbol{G}_{A_j}^{n+1} + \gamma \widehat{\boldsymbol{G}}_{A_j}^{n+1}$, and

$$\boldsymbol{G}_{A_j}^{n+1} = \boldsymbol{G}_{A_j}^n + \boldsymbol{\Gamma}_{A_j}^{n\mathsf{T}} \boldsymbol{\Gamma}_{A_j}^n, \qquad \boldsymbol{p}_{A_j}^{n+1} = \boldsymbol{p}_{A_j}^n + \boldsymbol{\Gamma}_{A_j}^{n\mathsf{T}} \boldsymbol{\Lambda}_{A_j}^n, \tag{7}$$

$$\widehat{\boldsymbol{G}}_{A_j}^{n+1} = \eta \widehat{\boldsymbol{G}}_{A_j}^n + \widehat{\boldsymbol{\Gamma}}_{A_j}^{n\mathsf{T}} \widehat{\boldsymbol{\Gamma}}_{A_j}^n, \qquad \hat{\boldsymbol{p}}_{A_j}^{n+1} = \eta \hat{\boldsymbol{p}}_{A_j}^n + \widehat{\boldsymbol{\Gamma}}_{A_j}^{n\mathsf{T}} \widehat{\boldsymbol{\Lambda}}_{A_j}^n. \tag{8}$$

Using the parameter $\tilde{\boldsymbol{\theta}}_{A_j}^{n+1}$, we obtain the *sketched version of imputed reward* as $\tilde{r}_{n,b}(A_j) := \langle \tilde{\boldsymbol{\theta}}_{A_j}^n, \boldsymbol{s}_{n,b} \rangle$ at step $b \in [\widehat{N}_j^n]$. Finally, we specify that the sketch submatrices $\{\boldsymbol{C}_A^n\}_{A \in \mathcal{A}, n \in [N]}$ and $\{\widehat{\boldsymbol{C}}_{A_j}^n\}_{A \in \mathcal{A}, n \in [N]}$ are the block construction of Sparser Johnson-Lindenstrauss Transform (SJLT) (Kane & Nelson, 2014), where the sketch size $c$ is divisible by the number of blocks $D$[1]. As shown in the last 4 columns in Table 1, sketching reduces the time complexity from $O(MBd^2)$ to $O(Mcd^2)$ for reward imputation of all $M$ actions in one episode, where $c < B$. When $Mc \approx B$, the overall time complexity of our reward imputation using sketching is even comparable to that without reward imputation which has a $O(Bd^2)$ time complexity.

**Updated Policy using Imputed Rewards.** Inspired by the UCB strategy (Li et al., 2010), the updated policy for online decision of the $(n+1)$-th episode can be formulated using the imputed rewards (parameterized by $\bar{\boldsymbol{\theta}}_A^{n+1}$ in Eq.(2)) or the sketched version of imputed rewards (parameterized by $\tilde{\boldsymbol{\theta}}_A^{n+1}$ in Eq.(6)). Specifically, for a new context $\boldsymbol{s}$,

*origin policy* $\bar{p}_{n+1}$ selects the action following $A \leftarrow \underset{A \in \mathcal{A}}{\arg\max} \langle \bar{\boldsymbol{\theta}}_A^{n+1}, \boldsymbol{s} \rangle + \omega [\boldsymbol{s}^\mathsf{T} (\boldsymbol{\Psi}_A^{n+1})^{-1} \boldsymbol{s}]^{\frac{1}{2}}$,

*sketched policy* $\tilde{p}_{n+1}$ selects the action following $A \leftarrow \underset{A \in \mathcal{A}}{\arg\max} \langle \tilde{\boldsymbol{\theta}}_A^{n+1}, \boldsymbol{s} \rangle + \alpha [\boldsymbol{s}^\mathsf{T} (\boldsymbol{W}_A^{n+1})^{-1} \boldsymbol{s}]^{\frac{1}{2}}$,

where $\omega \geq 0$ and $\alpha \geq 0$ are the regularization parameters in policy and their theoretical values are given in Theorem 4. We summarize the reward imputation using sketching and the sketched policy into Algorithm 2, called SPUIR. Similarly, we call the updating of the original policy that uses reward imputation without sketching, the Policy Updating with Imputed Rewards (PUIR).

---

[1] Since we set the number of blocks of SJLT as $D < d$, we omit $D$ in the complexity analysis.

---

**Algorithm 2** Sketched Policy Updating with Imputed Rewards (SPUIR) in the $(n+1)$-th episode

---

**INPUT:** Policy $\tilde{p}_n$, $\mathcal{D}_{n+1}$, $\mathcal{A} = \{A_j\}_{j \in [M]}$, $\alpha \geq 0, \eta \in (0,1), \gamma \in [0,1], \lambda > 0, \boldsymbol{W}_{A_j}^0 = \lambda \boldsymbol{I}_d, \boldsymbol{G}_{A_j}^0 = \widehat{\boldsymbol{G}}_{A_j}^0 = \boldsymbol{O}_d, \boldsymbol{p}_{A_j}^0 = \hat{\boldsymbol{p}}_{A_j}^0 = \boldsymbol{0}, \tilde{\boldsymbol{\theta}}_{A_j}^0 = \boldsymbol{0}, j \in [M]$, batch size $B$, sketch size $c$, number of block $D$

**OUTPUT:** Updated policy $\tilde{p}_{n+1}$

1: For all $j \in [M]$, store context vectors and rewards corresponding to the steps in which the action $A_j$ is executed, into $\boldsymbol{\Gamma}_{A_j}^n \in \mathbb{R}^{N_j^n \times d}$ and $\boldsymbol{\Lambda}_{A_j}^n \in \mathbb{R}^{N_j^n}$

2: For all $j \in [M]$, store context vectors corresponding to the steps in which the action $A_j$ is not executed into $\widehat{\boldsymbol{\Gamma}}_{A_j}^n \in \mathbb{R}^{\widehat{N}_j^n \times d}$, where $\widehat{N}_j^n \leftarrow B - N_j^n$

3: $\tilde{r}_{n,b}(A_j) \leftarrow \langle \tilde{\boldsymbol{\theta}}_{A_j}^n, \boldsymbol{s}_{n,b} \rangle$, for all $A_j \in \mathcal{A}$ and $b \in [\widehat{N}_j^n]$, where $\boldsymbol{s}_{n,b}$ is the $b$-th row of $\widehat{\boldsymbol{\Gamma}}_{A_j}^n$

4: Compute imputed reward vector $\widehat{\boldsymbol{R}}_{A_j}^n \leftarrow \{\tilde{r}_{n,1}(A_j), \ldots, \tilde{r}_{n,\widehat{N}_j^n}(A_j)\} \in \mathbb{R}^{\widehat{N}_j^n}$ for any $j \in [M]$

5: **for all** action $A_j \in \mathcal{A}$ **do**

6: $\quad \boldsymbol{G}_{A_j}^{n+1} \leftarrow \boldsymbol{G}_{A_j}^n + \boldsymbol{\Gamma}_{A_j}^{n\intercal}\boldsymbol{\Gamma}_{A_j}^n, \quad \boldsymbol{p}_{A_j}^{n+1} \leftarrow \boldsymbol{p}_{A_j}^n + \boldsymbol{\Gamma}_{A_j}^{n\intercal}\boldsymbol{\Lambda}_{A_j}^n \quad$ {Eq.(7)}

7: $\quad \widehat{\boldsymbol{G}}_{A_j}^{n+1} \leftarrow \eta\widehat{\boldsymbol{G}}_{A_j}^n + \widehat{\boldsymbol{\Gamma}}_{A_j}^{n\intercal}\widehat{\boldsymbol{\Gamma}}_{A_j}^n, \quad \hat{\boldsymbol{p}}_{A_j}^{n+1} \leftarrow \eta\hat{\boldsymbol{p}}_{A_j}^n + \widehat{\boldsymbol{\Gamma}}_{A_j}^{n\intercal}\widehat{\boldsymbol{\Lambda}}_{A_j}^n \quad$ {Eq.(8)}

8: $\quad \boldsymbol{W}_{A_j}^{n+1} \leftarrow \lambda\boldsymbol{I}_d + \boldsymbol{G}_{A_j}^{n+1} + \gamma\widehat{\boldsymbol{G}}_{A_j}^{n+1}, \quad \tilde{\boldsymbol{\theta}}_{A_j}^{n+1} \leftarrow (\boldsymbol{W}_{A_j}^{n+1})^{-1}(\boldsymbol{p}_{A_j}^{n+1} + \gamma\hat{\boldsymbol{p}}_{A_j}^{n+1}) \quad$ {Eq.(6)}

9: **end for**

10: $\tilde{p}_{n+1}(\boldsymbol{s})$ selects action $A \leftarrow \arg\max_{A \in \mathcal{A}} \langle \tilde{\boldsymbol{\theta}}_A^{n+1}, \boldsymbol{s} \rangle + \alpha[\boldsymbol{s}^\intercal (\boldsymbol{W}_A^{n+1})^{-1} \boldsymbol{s}]^{\frac{1}{2}}$ for a new context $\boldsymbol{s}$

11: **return** $\{\tilde{\boldsymbol{\theta}}_A^{n+1}\}_{A \in \mathcal{A}}, \{(\boldsymbol{W}_A^{n+1})^{-1}\}_{A \in \mathcal{A}}$

---

## 4 THEORETICAL ANALYSIS

We provide the instantaneous regret bound, prove the approximation error of sketching, and analyze the regret in the CBB setting. The detailed proofs can be found in Appendix B. We first demonstrate the instantaneous regret bound of the original solution $\bar{\boldsymbol{\theta}}_A^n$ in Eq.(1).

**Theorem 2** (Instantaneous Regret Bound). *Let $\eta \in (0,1)$ be the discount parameter, $\gamma \in [0,1]$ the imputation rate. In the $n$-th episode, if the rewards $\{R_{n,b}\}_{b \in [B]}$ are independent[2] and bounded by $C_R$, then, for any $b \in [B]$ and $\forall A \in \mathcal{A}$, with probability at least $1 - \delta$,*

$$\left| \langle \bar{\boldsymbol{\theta}}_A^n, \boldsymbol{s}_{n,b} \rangle - \langle \boldsymbol{\theta}_A^*, \boldsymbol{s}_{n,b} \rangle \right| \leq \left[ \lambda \|\boldsymbol{\theta}_A^*\|_2 + \nu + \gamma^{\frac{1}{2}}\eta^{-\frac{1}{2}}C_{\text{Imp}} \right] [\boldsymbol{s}_{n,b}^\intercal (\boldsymbol{\Psi}_A^n)^{-1} \boldsymbol{s}_{n,b}]^{\frac{1}{2}}, \quad (9)$$

*where $\boldsymbol{\Psi}_A^n = \lambda\boldsymbol{I}_d + \boldsymbol{\Phi}_A^n + \gamma\widehat{\boldsymbol{\Phi}}_A^n$, $\nu = [2C_R^2 \log(2MB/\delta)]^{\frac{1}{2}}$, and $C_{\text{Imp}} > 0$. The first term on the right-hand side of Eq.(9) can be seen as the bias term for the reward imputation, while the second term is the variance term. The variance term of our algorithm is not larger than that without the reward imputation, i.e, for any $\boldsymbol{s} \in \mathbb{R}^d$, $[\boldsymbol{s}^\intercal (\boldsymbol{\Psi}_A^n)^{-1} \boldsymbol{s}]^{\frac{1}{2}} \leq [\boldsymbol{s}^\intercal (\lambda\boldsymbol{I}_d + \boldsymbol{\Phi}_A^n)^{-1} \boldsymbol{s}]^{\frac{1}{2}}$. Further, a larger imputation rate $\gamma$ leads to a smaller variance term $[\boldsymbol{s}^\intercal (\boldsymbol{\Psi}_A^n)^{-1} \boldsymbol{s}]^{\frac{1}{2}}$.*

**Remark 1** (Smaller Variance). *From Theorem 2, we can observe that the proposed reward imputation achieves a smaller variance ($[\boldsymbol{s}_{n,b}^\intercal (\boldsymbol{\Psi}_A^n)^{-1} \boldsymbol{s}_{n,b}]^{\frac{1}{2}}$) than that without the reward imputation.*

**Remark 2** (Controllable Bias). *Our reward imputation approach incurs a bias term $\gamma^{\frac{1}{2}}\eta^{-\frac{1}{2}}C_{\text{Imp}}$ in addition to the two bias terms $\lambda\|\boldsymbol{\theta}_A^*\|_2, \nu$ that exist in every UCB-based policy. But this additional bias term is controllable due to the presence of imputation rate $\gamma$ that can help controlling the additional bias. Moreover, the term $C_{\text{Imp}}$ in the additional bias can be replaced by a function $f_{\text{Imp}}(n)$, and $f_{\text{Imp}}(n)$ is monotonic decreasing w.r.t. number of episodes $n$ provided that the mild condition $\sqrt{\eta} = \Theta(d^{-1})$ holds (the definition and analysis about $f_{\text{Imp}}$ can be found in Appendix B.1). Overall, the imputation rate $\gamma$ controls a trade-off between the bias term and the variance term, and we will design a rate-scheduled approach for choosing the imputation rate $\gamma$ in Section 5.*

Although some approximation error bounds using SJLT have been proposed (Nelson & Nguyên, 2013; Kane & Nelson, 2014; Zhang & Liao, 2019), it is still unknown what is the lower bound of the sketch size while applying SJLT to the sketched ridge regression problem in our SPUIR. Next, we prove the approximation error as well as the lower bound of the sketch size for the sketched ridge regression problem. For convenience, we drop all the superscripts and subscripts in this result.

**Theorem 3** (Approximation Error Bound of Imputation using Sketching). *Denote the imputation regularized ridge regression function by $F(\boldsymbol{\theta})$ (defined in Eq.(1)) and the sketched ridge regression*

---

[2]The assumption about conditional independence of the rewards is commonly used in the bandits literature, which can be ensured using a master technology as a theoretical construction (Auer, 2002; Chu et al., 2011).

function by $F^{\mathrm{S}}(\boldsymbol{\theta})$ (defined in Eq.(5)) for reward imputation, whose solutions (i.e., the estimated reward parameter vectors) are $\bar{\boldsymbol{\theta}} = \arg\min_{\boldsymbol{\theta}\in\mathbb{R}^d} F(\boldsymbol{\theta})$ and $\tilde{\boldsymbol{\theta}} = \arg\min_{\boldsymbol{\theta}\in\mathbb{R}^d} F^{\mathrm{S}}(\boldsymbol{\theta})$. Let $\gamma \in [0,1]$ be the imputation rate, $\lambda > 0$ the regularization parameter, $\delta \in (0,0.1]$, $\varepsilon \in (0,1)$, $\boldsymbol{L}_{\mathrm{all}} = [\boldsymbol{L}; \sqrt{\gamma}\widehat{\boldsymbol{L}}]$, and $\rho_\lambda = \|\boldsymbol{L}_{\mathrm{all}}\|_2^2/(\|\boldsymbol{L}_{\mathrm{all}}\|_2^2 + \lambda)$. If $\boldsymbol{\Pi}$ and $\widehat{\boldsymbol{\Pi}}$ are SJLT, assuming that $D = \Theta(\varepsilon^{-1}\log^3(d\delta^{-1}))$ and the sketch size $c = \Omega\left(d\,\mathrm{polylog}\left(d\delta^{-1}\right)/\varepsilon^2\right)$, with probability at least $1 - \delta$, $F(\tilde{\boldsymbol{\theta}}) \le (1 + \rho_\lambda\varepsilon)F(\bar{\boldsymbol{\theta}})$ and $\|\tilde{\boldsymbol{\theta}} - \bar{\boldsymbol{\theta}}\|_2 = O\left(\sqrt{\rho_\lambda\varepsilon}\right)$ hold.

To measure the convergence of approximating the optimal policy in sequential decision-making, we define the *regret* of SPUIR against the optimal policy as follows:

$$\mathrm{Reg}\left(\{A_{I_{n,b}}\}_{n\in[N],b\in[B]}\right) := \max_{A\in\mathcal{A}} \sum_{n\in[N],b\in[B]}[\langle\boldsymbol{\theta}_A^*, \boldsymbol{s}_{n,b}\rangle - \langle\boldsymbol{\theta}_{A_{I_{n,b}}}^*, \boldsymbol{s}_{n,b}\rangle],$$

where $I_{n,b}$ denotes the index of the executed action using the sketched policy $\tilde{p}_n$ (parameterized by $\{\tilde{\boldsymbol{\theta}}_A^n\}_{A\in\mathcal{A}}$) at step $b$ in the $n$-th episode. Finally, we demonstrate the regret bound of SPUIR.

**Theorem 4** (Regret Bound of SPUIR). *Let $T = BN$ be the overall number of steps, $\eta \in (0,1)$ be the discount parameter, $\gamma \in [0,1]$ the imputation rate, $\lambda > 0$ the regularization parameter, $C_{\boldsymbol{\theta}^*}^{\max} = \max_{A\in\mathcal{A}} \|\boldsymbol{\theta}_A^*\|_2$, $C_{\mathrm{Imp}}$ be a positive constant. Assume that the conditional independence assumption in Theorem 2 holds and the upper bound of rewards is $C_R$, $M = O(\mathrm{poly}(d))$, $T \ge d^2$, $\nu = [2C_R^2\log(2MB/\delta_1)]^{\frac{1}{2}}$ with $\delta_1 \in (0,1)$,*

$$\omega = \lambda C_{\boldsymbol{\theta}^*}^{\max} + \nu + \gamma^{\frac{1}{2}}\eta^{-\frac{1}{2}}C_{\mathrm{Imp}}, \quad \alpha = \omega C_\alpha,$$

*where $C_\alpha > 0$ which decreases with increase of $1/\varepsilon$ and $\varepsilon \in (0,1)$. Let $\delta_2 \in (0,0.1]$, $\rho_\lambda < 1$ be the constant defined in Theorem 3, and $C_{\mathrm{reg}}$ be a positive constant that decreases with increase of $1/\varepsilon$. For the sketch matrices $\{\boldsymbol{\Pi}_A^n\}_{A\in\mathcal{A},n\in[N]}$ and $\{\widehat{\boldsymbol{\Pi}}\}_{A\in\mathcal{A},n\in[N]}$, assuming that the number of blocks in SJLT $D = \Theta(\varepsilon^{-1}\log^3(d\delta_2^{-1}))$, and the sketch size satisfying $c = \Omega\left(d\,\mathrm{polylog}\left(d\delta_2^{-1}\right)/\varepsilon^2\right)$, then, for an arbitrary sequence of contexts $\{\boldsymbol{s}_{n,b}\}_{n\in[N],b\in[B]}$, with probability at least $1 - N(\delta_1 + \delta_2)$,*

$$\mathrm{Reg}(\{A_{I_{n,b}}\}_{n\in[N],b\in[B]}) \le 2\alpha C_{\mathrm{reg}}\sqrt{10M}\log(T+1)(\sqrt{dT} + dB) + O\left(T\sqrt{\rho_\lambda\epsilon d}/B\right). \quad (10)$$

**Remark 3.** *Setting $B = O(\sqrt{T/d})$ and $\rho_\lambda\epsilon = 1/d$ yields a sublinear upper bound of regret of order $\widetilde{O}(\sqrt{MdT})$[3] provided that the sketch size $c = \Omega(\rho_\lambda^2 d^3\,\mathrm{polylog}(d\delta_2^{-1}))$. We can observe that the lower bound of $c$ is independent of the overall number of steps $T$, and a theoretical value of the batch size is $B = C_B\sqrt{T/d} = C_B^2 N/d$, where setting $C_B \approx 25$ is a suitable choice that has been verified in the experiments in Section 6. In particular, when $\rho_\lambda = O(1/d)$, the sketch size of order $c = \Omega(d\,\mathrm{polylog}d)$ is sufficient to achieve a sublinear regret, which has been demonstrated in our experimental results. Since the lower bound of regret for contextual batched bandit in (Han et al., 2020) assumes that there are only two actions and both the actions share the same true reward model, it can not be applied to our CBB setting where each action corresponds to a different reward model. Despite the lack of the lower bound in CBB setting, from the theoretical results of regret, we can observe that our SPUIR admits several advantages: (a) The order of our regret bound is not higher than those in the literature of contextual bandits in the fully-online setting (i.e., $B = 1$) (Li et al., 2019; Dimakopoulou et al., 2019), which is a more simple setting than ours; (b) The first term in the regret bound Eq.(10) measures the performance of policy updating using imputed rewards (called "policy error"). From Theorem 2 and Remark 1&2, we obtain that, in each episode, our policy updating has a smaller variance than the policy without the reward imputation, and incurs a decreasing additional bias under mild conditions, leading to a tighter regret (i.e., smaller policy error) after some number of episodes. (c) The second term on the right-hand side of Eq.(10) is of order $O(T\sqrt{\rho_\lambda\epsilon d}/B)$, which is incurred by the sketching approximation using SJLT (called "sketching error"). This sketching error does not have any influence on the order of regret of SPUIR, which may even have a lower order with a suitable choice of $\rho_\lambda\varepsilon$, e.g., setting $\rho_\lambda\varepsilon = T^{-1/4}d^{-1}$ yields a sketching error of order $O(T^{3/8}d^{1/2})$ provided that $c = \Omega(\rho_\lambda^2 d^3\,\mathrm{polylog}(d\delta_2^{-1})\sqrt{T})$.*

## 5 Extensions of Our Approach

To make the proposed reward imputation approach more feasible and practical, we tackle the following two research questions by designing variants of our approach following the theoretical results:

---

[3]We use the notation of $\widetilde{O}$ to suppress logarithmic factors in the overall number of steps $T$.

**RQ1 (Parameter Selection):** *Can we set the imputation rate $\gamma$ without tuning?*

**RQ2 (Nonlinear Reward):** *Can we apply the proposed reward imputation approach to the case where the expectation of true rewards is nonlinear?*

**Rate-Scheduled Approach.** For RQ1, we equip PUIR and SPUIR with a rate-scheduled approach, called PUIR-RS and SPUIR-RS, respectively. From Remark 1&2, a larger imputation rate $\gamma$ leads to a smaller variance while increasing the bias, while the bias term includes a monotonic decreasing function w.r.t. number of episodes under mild conditions. Therefore, we can gradually increase $\gamma$ with the number of episodes, avoiding the large bias at the beginning of reward imputation. Specifically, we set $\gamma = X\%$ for episodes from $(X - 10)\% \times N$ to $X\% \times N$, where $X \in [10, 100]$.

**Application to Nonlinear Rewards.** For RQ2, we provide nonlinear versions of reward imputation. We use linearization technologies of nonlinear rewards, rather than directly setting the rewards as nonlinear functions (Valko et al., 2013; Chatterji et al., 2019), avoiding the linear regret or curse of kernelization. Specifically, instead of using the linear imputed reward $\tilde{r}_{n,b}(A_j) := \langle \tilde{\boldsymbol{\theta}}_{A_j}^n, \boldsymbol{s}_{n,b} \rangle$, we use the following linearized nonlinear imputed rewards, denotes by $\mathcal{T}_{n,b}(\boldsymbol{\theta}, A)$:

**1) SPUIR-Exp.** We assume that the expected reward is an exponential function as $G_{\mathrm{E}}(\boldsymbol{\theta}, \boldsymbol{s}) = \exp\left(\boldsymbol{\theta}^{\intercal} \boldsymbol{s}\right)$. Then $\mathcal{T}_{n,b}(\boldsymbol{\theta}, A) = \langle \boldsymbol{\theta}, \nabla_{\boldsymbol{\theta}} G_{\mathrm{E}}(\boldsymbol{\theta}, \boldsymbol{s}_{n,b}) \rangle$, where $\nabla_{\boldsymbol{\theta}} G_{\mathrm{E}}(\boldsymbol{\theta}, \boldsymbol{s}_{n,b}) = \exp\left(\boldsymbol{\theta}^{\intercal} \boldsymbol{s}_{n,b}\right) \boldsymbol{s}_{n,b}$.

**2) SPUIR-Poly.** When the expected reward is a polynomial function as $G_{\mathrm{P}}(\boldsymbol{\theta}, \boldsymbol{s}) = \left(\boldsymbol{\theta}^{\intercal} \boldsymbol{s}\right)^2$. Then $\mathcal{T}_{n,b}(\boldsymbol{\theta}, A) = \langle \boldsymbol{\theta}, \nabla_{\boldsymbol{\theta}} G_{\mathrm{P}}(\boldsymbol{\theta}, \boldsymbol{s}_{n,b}) \rangle$, where $\nabla_{\boldsymbol{\theta}} G_{\mathrm{P}}(\boldsymbol{\theta}, \boldsymbol{s}_{n,b}) = 2\left(\boldsymbol{\theta}^{\intercal} \boldsymbol{s}_{n,b}\right) \boldsymbol{s}_{n,b}$.

**3) SPUIR-Kernel.** Consider that the underlying expected reward in a Gaussian reproducing kernel Hilbert space (RKHS). We use $\mathcal{T}_{n,b}(\boldsymbol{\theta}, A) = \langle \boldsymbol{\theta}, \phi(\boldsymbol{s}_{n,b}) \rangle$ in random feature space, where the random feature mapping $\phi$ can be explicitly defined as in (Rahimi & Recht, 2007).

For SPUIR-Exp and SPUIR-Poly, combining the linearization of convex functions (Shalev-Shwartz, 2011) with Theorem 4 yields the regret bounds of the same order. For SPUIR-Kernel, using the approximation error of random features (Rahimi & Recht, 2008), we can obtain a regret bound with an additional error of order $O(B/\sqrt{d_{\mathrm{r}}})$, where $d_{\mathrm{r}}$ is the dimension of the random features.

# 6 EXPERIMENTS

We empirically evaluated the performance of our algorithms on 3 datasets: the synthetic dataset, publicly available Criteo dataset[4] (`Criteo-recent`, `Criteo-all`), and dataset collected from a real commercial app for coupon recommendation (`commercial product`).

**Experimental Settings.** We compared our algorithms with: Sequential Batch UCB (SBUCB) (Han et al., 2020), Batched linear EXP3 (BEXP3) (Neu & Olkhovskaya, 2020), Batched linear EXP3 using Inverse Propensity Weighting (BEXP3-IPW) (Bistritz et al., 2019), Batched Balanced Linear Thompson Sampling (BLTS-B) (Dimakopoulou et al., 2019), and Sequential version of Delayed Feedback Model (DFM-S) (Chapelle, 2014). We implemented all the algorithms on Intel(R) Xeon(R) Silver 4114 CPU@2.20GHz, and repeated the experiments 20 times. We tested the performance of algorithms in streaming recommendation scenarios, where the reward is represented by a linear combination of the click and conversion behaviors of users. According to Remark 3, we set the batch size as $B = C_B^2 N/d$, the constant $C_B \approx 25$, and the sketch size $c = 150$ on all the datasets. The average reward was used to evaluate the accuracy of algorithms.

**Performance Evaluation.** Figure 3(a)–(c) reports the average reward of SPUIR with its variants and the baselines. We observed that SPUIR and its variants achieved higher average rewards, demonstrating the effectiveness of our reward imputation. Moreover, SPUIR and its rate-scheduled version SPUIR-RS had similar performances compared with the origin PUIR, which indicates the practical effectiveness of our variants and verifies the correctness of the theoretical analyses. The results on `commercial product` in Table 2 indicate that SPUIR outperformed the second-best baseline with the improvements of 1.07% CVR (conversion rate) and 1.12% CTCVR (post-view click-through&conversion rate). Besides, our reward imputation approaches were more efficient than DFM-S, BLTS-B. The variants using sketching of our algorithms (SPUIR, SPUIR-RS) significantly reduced the time costs of reward imputation, and took less than twice as long to run compared to the baselines without reward imputation (SBUCB, BEXP3, BEXP3-IPW). Figure 3(d) illustrates performances of SPUIR and its nonlinear variants, where SPUIR-Kernel achieved the highest rewards indicating the effectiveness of the nonlinear generalization of our approach. For different decision tasks, a suitable nonlinear reward model needs to be selected for better performances.

**Parameter Influence.** From the regret bound Eq.(10), we can observe that a larger batch size $B$ results in a larger first term (of order $O(B)$, called policy error) but a smaller second term (of order

---

[4]https://labs.criteo.com/2013/12/conversion-logs-dataset/

Table 2: Performance comparison of coupon recommendation on `commercial product`

| Algorithm | CVR (mean $\pm$ std) | CTCVR (mean $\pm$ std) | Time (sec., mean $\pm$ std) |
|---|---|---|---|
| DFM-S | $0.8656 \pm 0.0473$ | $0.3317 \pm 0.0218$ | $302.3140 \pm 8.3045$ |
| SBUCB | $0.8569 \pm 0.0037$ | $0.4277 \pm 0.0084$ | $43.5435 \pm 0.3659$ |
| BEXP3 | $0.4846 \pm 0.0205$ | $0.2425 \pm 0.0116$ | $53.5001 \pm 0.9220$ |
| BEXP3-IPW | $0.4862 \pm 0.0187$ | $0.2436 \pm 0.0113$ | $56.0101 \pm 1.4142$ |
| BLTS-B | $0.8663 \pm 0.0178$ | $0.4285 \pm 0.0157$ | $218.2109 \pm 1.8198$ |
| PUIR | $0.8807 \pm 0.0053$ | $0.4411 \pm 0.0029$ | $184.3575 \pm 2.2346$ |
| SPUIR | $0.8770 \pm 0.0059$ | $0.4397 \pm 0.0032$ | $81.5753 \pm 1.5879$ |
| PUIR-RS | $0.8763 \pm 0.0056$ | $0.4389 \pm 0.0030$ | $180.4999 \pm 1.7763$ |
| SPUIR-RS | $0.8758 \pm 0.0058$ | $0.4391 \pm 0.0031$ | $80.8003 \pm 2.9030$ |

Figure 3: (a), (b), (c): Average rewards of the compared algorithms, the proposed SPUIR and its variants on synthetic dataset, Criteo dataset, and the real commercial product data, where we omitted the curves of algorithms whose average rewards are $5\%$ lower than the highest reward; (d): SPUIR and its three nonlinear variants on synthetic dataset; (e): SPUIR with different batch sizes on `Criteo-recent`; (f): SPUIR and SPUIR-RS with different sketch sizes on synthetic dataset

$O(1/B)$, called sketching error), indicating that a suitable batch size $B$ needs to be set. This conclusion was empirically verified in Figure 3(e), where $B = 1,000$ ($C_B = 25$) yields better empirical performance in terms of the average reward. Similar phenomenon can also be observed on Criteo dataset and `commercial product`. All of the results verified the theoretical results in Remark 3: $B = C_B\sqrt{T/d} = C_B^2 N/d$ is a suitable choice while setting $C_B \approx 25$. From the results in Figure 3(f) we observe that, for our SPUIR and SPUIR-RS, the performances significantly increased when the sketch size $c$ reached $10\%B$ ($\approx d\log d$), which demonstrates the conclusion in Remark 3 that only the sketch size of order $c = \Omega(d\, \mathrm{polylog} d)$ is needed for satisfactory performance.

## 7 CONCLUSION

Partial-information feedback is ubiquitous in real-world applications, where reward feedback is usually underutilized for learning. This paper proposes a theoretically sound and computationally efficient reward imputation approach for contextual batched bandits, which mimics the reward generation mechanism of the environment approximating the setting of full-information feedback. The proposed reward imputation approach reduces the time complexity of imputation on large batches of data using sketching, achieves a relative-error bound for sketching approximation, has an instantaneous regret with a controllable bias and a smaller variance, and enjoys a sublinear regret bound against the optimal policy. The theoretical formulation and algorithmic implementation may provide an efficient reward imputation scheme for online learning under limited feedback.

## ETHICS STATEMENT

To verify the effectiveness and efficiency of our algorithms on real products, we conducted experiments on a real dataset collected from a commercial social app. We call this dataset `commercial product`, where the data were collected after the users gave consent, and did not contain any personally identifiable information or offensive content.

## REPRODUCIBILITY STATEMENT

The source code of the proposed algorithms is submitted as supplementary materials. For theoretical results, clear explanations of any assumptions and a complete proof of the claims are included as an appendix. The detailed experimental settings are also provided in the appendix. Since the dataset `commercial product` is non-public, we did not provide a URL. We will release this non-public dataset after the publication of this paper, and the link to download this non-public dataset is to be included in the final paper.

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

## A    DETAILED PROBLEM FORMULATION AND DETAILED PROOF IN PROBLEM ANALYSIS

In this part, we give the detailed problem formulation and the detailed proof of Theorem 1 in problem analysis in Section 2.

### A.1    DETAILED PROBLEM FORMULATION OF CBB

In this paper, we focus on the setting of contextual batched bandits (CBB), which can be formulated as a 6-tuple $\langle \mathcal{S}, \mathcal{A}, p, R, N, B \rangle$:

**Context space** $\mathcal{S} \subseteq \mathbb{R}^d$ means a vector space containing the context information received at each step, e.g., context summarizes the information of both the user and items in recommendation scenarios.

**Action space** $\mathcal{A} = \{A_j\}_{j \in [M]}$ contains $M$ candidate actions for execution. As an example, in recommender systems, each action corresponds to a candidate item, and selecting an action means that the corresponding item is recommended.

**Policy** $p$ determines which action to take at each step, which is a function of the context $s \in \mathcal{S}$ and outputs an action for execution (or a selection distribution over action space $\mathcal{A}$).

**Reward** $R$ in CBB is a *partial-information feedback* where rewards are unobserved for the non-executed actions. Consider a stochastic bandit setting, where the expectation of the true reward is assumed to be a function of the context $s \in \mathcal{S}$. In particular, different from the shared expectation function of true rewards in existing batch bandits (Han et al., 2020), we assume that the expectation functions of true rewards are different for each action, where each expectation function corresponds to an unknown parameter vector $\boldsymbol{\theta}_A^* \in \mathbb{R}^d$, $A \in \mathcal{A}$. This setting for rewards matches many real-world applications, e.g., each action corresponds to a different category of candidate coupons in coupon recommendation.

**Number of episodes** $N$. The decision process in CBB is partitioned into $N$ episodes. Within one episode, the agent updates the policy using the collected data, and then interacts with the environment for multiple steps using the updated and fixed policy.

**Batch size** $B$ is the number of steps in each episode. That is, in each episode, the agent interacts with the environment $B$ times using a fixed policy, and stores the contexts, executed actions, and observed rewards into a data buffer $\mathcal{D}$ at the end of each episode.

### A.2    DETAILED DESCRIPTION AND PROOF OF THEOREM 1 IN PROBLEM ANALYSIS

We present some theoretical findings about the regret difference between the partial-information feedback and the full-information feedback. Assuming that the agent in the CBB setting can observe the rewards of all the candidate actions from the environment at each step, we apply the batched UCB policy (Han et al., 2020) to this setting (see Algorithm 3), We demonstrate an instantaneous regret bound in Theorem 5, where Theorem 5 is a detailed version of Theorem 1 in Section 2.

---

**Algorithm 3** Batch UCB Policy Updating in the $(n+1)$-th episode in Full-Information CBB Setting

---

**INPUT:** Policy $p_n$, data buffer $\mathcal{D}_{n+1}$, action space $\mathcal{A} = \{A_j\}_{j \in [M]}$, $\boldsymbol{\theta}_{A_j}^0 = \mathbf{0}$, $j \in [M]$, batch size $B$

**OUTPUT:** Updated policy $p_{n+1}$

1: Let $\widetilde{\boldsymbol{L}}_n \in \mathbb{R}^{(n+1)B \times d}$ be the matrix that stores all the context vectors till the $n$-th episode as the row vectors

2: For $\forall A \in \mathcal{A}$, let $\widetilde{\boldsymbol{T}}_A^n \in \mathbb{R}^{(n+1)B}$ be the reward vector that stores all the rewards of action $A \in \mathcal{A}$ till the $n$-th episode

3: // Policy Updating

4: $\boldsymbol{\Upsilon}_{n+1} \leftarrow \widetilde{\boldsymbol{L}}_n^\mathsf{T} \widetilde{\boldsymbol{L}}_n$

5: **for all** action $A \in \mathcal{A}$ **do**

6: $\quad \boldsymbol{\theta}_A^{n+1} \leftarrow (\boldsymbol{I}_d + \boldsymbol{\Upsilon}_{n+1})^{-1} \widetilde{\boldsymbol{L}}_n^\mathsf{T} \widetilde{\boldsymbol{T}}_A^n$

7: **end for**

8: For a new context $s$, $p_{n+1}(s)$ is to choose the action following: $A \leftarrow \underset{A \in \mathcal{A}}{\arg\max} \left\langle \boldsymbol{\theta}_A^{n+1}, s \right\rangle$

9: **return** $\left\{ \boldsymbol{\theta}_A^{n+1} \right\}_{A \in \mathcal{A}}$

---

**Theorem 5** (Instantaneous Regret Bound in Full-Information CBB Setting, Detailed Version of Theorem 1). *Let $\widetilde{\boldsymbol{L}}_{n-1} \in \mathbb{R}^{nB \times d}$ be the matrix that stores all the context vectors till the $(n-1)$-th episode as the row vectors, and $\widetilde{\boldsymbol{T}}_A^{n-1} \in \mathbb{R}^{nB}$ be the reward vector that stores all the rewards of action $A \in \mathcal{A}$ till the $(n-1)$-th episode. Given the action space $\mathcal{A} = \{A_j\}_{j \in [M]}$, in the $n$-th episode, assume that the rewards are independent and bounded by $C_R$. Then, with probability at least $1 - \delta$, for any $b \in [B]$ and $\forall A \in \mathcal{A}$, we have the following instantaneous regret bound in the $n$-th episode*

$$|\langle \boldsymbol{\theta}_A^n, \boldsymbol{s}_{n,b} \rangle - \langle \boldsymbol{\theta}_A^*, \boldsymbol{s}_{n,b} \rangle| \leq \left[ \|\boldsymbol{\theta}_A^*\|_2 + \sqrt{2C_R^2 \log(2MB/\delta)} \right] \sqrt{\boldsymbol{s}_{n,b}^\intercal (\boldsymbol{I}_d + \boldsymbol{\Upsilon}_n)^{-1} \boldsymbol{s}_{n,b}}, \quad (11)$$

*where $\boldsymbol{\Upsilon}_n = \widetilde{\boldsymbol{L}}_{n-1}^\intercal \widetilde{\boldsymbol{L}}_{n-1}$ and the parameter of reward model $\boldsymbol{\theta}_A^n$ in the batched UCB policy is obtained by*

$$\boldsymbol{\theta}_A^n := \arg\min_{\boldsymbol{\theta} \in \mathbb{R}^d} \left\| \widetilde{\boldsymbol{L}}_{n-1} \boldsymbol{\theta} - \widetilde{\boldsymbol{T}}_A^{n-1} \right\|_2^2 + \|\boldsymbol{\theta}\|_2^2 = (\boldsymbol{I}_d + \boldsymbol{\Upsilon}_n)^{-1} \widetilde{\boldsymbol{L}}_{n-1}^\intercal \widetilde{\boldsymbol{T}}_A^{n-1}.$$

*Further, the instantaneous regret bound Eq.(11) in FI-CBB setting is tighter than that in CBB setting (i.e., using the partial-information feedback). In particular, the variance term $\sqrt{\boldsymbol{s}_{n,b}^\intercal (\boldsymbol{I}_d + \boldsymbol{\Upsilon}_n)^{-1} \boldsymbol{s}_{n,b}}$ is smaller than that in CBB setting.*

*Proof of Theorem 1.* By the formulation of $\boldsymbol{\theta}_A^n$ and the triangle inequality, we first obtain that

$$\begin{aligned}
&|\langle \boldsymbol{\theta}_A^n, \boldsymbol{s}_{n,b} \rangle - \langle \boldsymbol{\theta}_A^*, \boldsymbol{s}_{n,b} \rangle| \\
&= \left| \boldsymbol{s}_{n,b}^\intercal (\boldsymbol{I}_d + \boldsymbol{\Upsilon}_n)^{-1} \widetilde{\boldsymbol{L}}_{n-1}^\intercal \widetilde{\boldsymbol{T}}_A^{n-1} - \boldsymbol{s}_{n,b}^\intercal \boldsymbol{\theta}_A^* \right| \\
&= \left| \boldsymbol{s}_{n,b}^\intercal (\boldsymbol{I}_d + \boldsymbol{\Upsilon}_n)^{-1} \left[ \widetilde{\boldsymbol{L}}_{n-1}^\intercal \widetilde{\boldsymbol{T}}_A^{n-1} - (\boldsymbol{I}_d + \boldsymbol{\Upsilon}_n) \boldsymbol{\theta}_A^* \right] \right| \\
&= \left| \boldsymbol{s}_{n,b}^\intercal (\boldsymbol{I}_d + \boldsymbol{\Upsilon}_n)^{-1} \left[ \widetilde{\boldsymbol{L}}_{n-1}^\intercal \widetilde{\boldsymbol{T}}_A^{n-1} - \left( \boldsymbol{I}_d + \widetilde{\boldsymbol{L}}_{n-1}^\intercal \widetilde{\boldsymbol{L}}_{n-1} \right) \boldsymbol{\theta}_A^* \right] \right| \\
&= \left| \boldsymbol{s}_{n,b}^\intercal (\boldsymbol{I}_d + \boldsymbol{\Upsilon}_n)^{-1} \widetilde{\boldsymbol{L}}_{n-1}^\intercal \left( \widetilde{\boldsymbol{T}}_A^{n-1} - \widetilde{\boldsymbol{L}}_{n-1} \boldsymbol{\theta}_A^* \right) - \boldsymbol{s}_{n,b}^\intercal (\boldsymbol{I}_d + \boldsymbol{\Upsilon}_n)^{-1} \boldsymbol{\theta}_A^* \right| \\
&\leq \left| \boldsymbol{s}_{n,b}^\intercal (\boldsymbol{I}_d + \boldsymbol{\Upsilon}_n)^{-1} \widetilde{\boldsymbol{L}}_{n-1}^\intercal \left( \widetilde{\boldsymbol{T}}_A^{n-1} - \widetilde{\boldsymbol{L}}_{n-1} \boldsymbol{\theta}_A^* \right) \right| + \left| \boldsymbol{s}_{n,b}^\intercal (\boldsymbol{I}_d + \boldsymbol{\Upsilon}_n)^{-1} \boldsymbol{\theta}_A^* \right|
\end{aligned} \quad (12)$$

Next, we bound the two terms in the last row of Eq.(12).

**Bounding** $\left| \boldsymbol{s}_{n,b}^\intercal (\boldsymbol{I}_d + \boldsymbol{\Upsilon}_n)^{-1} \widetilde{\boldsymbol{L}}_{n-1}^\intercal \left( \widetilde{\boldsymbol{T}}_A^{n-1} - \widetilde{\boldsymbol{L}}_{n-1} \boldsymbol{\theta}_A^* \right) \right|$:

Since $\mathrm{E}\left[ \widetilde{\boldsymbol{T}}_A^{n-1} \right] = \widetilde{\boldsymbol{L}}_{n-1} \boldsymbol{\theta}_A^*$ and the received rewards are independent, by the Azuma-Hoeffding bound, we have

$$\begin{aligned}
&\Pr\left\{ \left| \boldsymbol{s}_{n,b}^\intercal (\boldsymbol{I}_d + \boldsymbol{\Upsilon}_n)^{-1} \widetilde{\boldsymbol{L}}_{n-1}^\intercal \left( \widetilde{\boldsymbol{T}}_A^{n-1} - \widetilde{\boldsymbol{L}}_{n-1} \boldsymbol{\theta}_A^* \right) \right| \geq \nu \sqrt{\boldsymbol{s}_{n,b}^\intercal (\boldsymbol{I}_d + \boldsymbol{\Upsilon}_n)^{-1} \boldsymbol{s}_{n,b}} \right\} \\
&\leq 2 \exp\left\{ -\frac{\nu^2 \boldsymbol{s}_{n,b}^\intercal (\boldsymbol{I}_d + \boldsymbol{\Upsilon}_n)^{-1} \boldsymbol{s}_{n,b}}{2 C_R^2 \| \widetilde{\boldsymbol{L}}_{n-1} (\boldsymbol{I}_d + \boldsymbol{\Upsilon}_n)^{-1} \boldsymbol{s}_{n,b} \|_2^2} \right\},
\end{aligned} \quad (13)$$

where $\nu > 0$ is some constant. Since

$$\begin{aligned}
\| \widetilde{\boldsymbol{L}}_{n-1} (\boldsymbol{I}_d + \boldsymbol{\Upsilon}_n)^{-1} \boldsymbol{s}_{n,b} \|_2^2 &= \boldsymbol{s}_{n,b}^\intercal (\boldsymbol{I}_d + \boldsymbol{\Upsilon}_n)^{-1} \widetilde{\boldsymbol{L}}_{n-1}^\intercal \widetilde{\boldsymbol{L}}_{n-1} (\boldsymbol{I}_d + \boldsymbol{\Upsilon}_n)^{-1} \boldsymbol{s}_{n,b} \\
&\leq \boldsymbol{s}_{n,b}^\intercal (\boldsymbol{I}_d + \boldsymbol{\Upsilon}_n)^{-1} \left( \boldsymbol{I}_d + \widetilde{\boldsymbol{L}}_{n-1}^\intercal \widetilde{\boldsymbol{L}}_{n-1} \right) (\boldsymbol{I}_d + \boldsymbol{\Upsilon}_n)^{-1} \boldsymbol{s}_{n,b} \\
&\leq \boldsymbol{s}_{n,b}^\intercal (\boldsymbol{I}_d + \boldsymbol{\Upsilon}_n)^{-1} (\boldsymbol{I}_d + \boldsymbol{\Upsilon}_n) (\boldsymbol{I}_d + \boldsymbol{\Upsilon}_n)^{-1} \boldsymbol{s}_{n,b} \\
&= \boldsymbol{s}_{n,b}^\intercal (\boldsymbol{I}_d + \boldsymbol{\Upsilon}_n)^{-1} \boldsymbol{s}_{n,b},
\end{aligned}$$

combing with Eq.(13) implies the following results

$$\begin{aligned}
&\Pr\left\{ \left| \boldsymbol{s}_{n,b}^\intercal (\boldsymbol{I}_d + \boldsymbol{\Upsilon}_n)^{-1} \widetilde{\boldsymbol{L}}_{n-1}^\intercal \left( \widetilde{\boldsymbol{T}}_A^{n-1} - \widetilde{\boldsymbol{L}}_{n-1} \boldsymbol{\theta}_A^* \right) \right| \geq \nu \sqrt{\boldsymbol{s}_{n,b}^\intercal (\boldsymbol{I}_d + \boldsymbol{\Upsilon}_n)^{-1} \boldsymbol{s}_{n,b}} \right\} \\
&\leq 2 \exp\left\{ -\frac{\nu^2}{2 C_R^2} \right\}.
\end{aligned} \quad (14)$$

Combing Eq.(14) with the union bound, yields that, with probability at least $1 - \delta$, for any $b \in [B]$ and $\forall A \in \mathcal{A}$,

$$\left| \boldsymbol{s}_{n,b}^{\mathsf{T}} \left( \boldsymbol{I}_d + \boldsymbol{\Upsilon}_n \right)^{-1} \widetilde{\boldsymbol{L}}_{n-1}^{\mathsf{T}} \left( \widetilde{\boldsymbol{T}}_A^{n-1} - \widetilde{\boldsymbol{L}}_{n-1} \boldsymbol{\theta}_A^* \right) \right| \leq \nu \sqrt{ \boldsymbol{s}_{n,b}^{\mathsf{T}} \left( \boldsymbol{I}_d + \boldsymbol{\Upsilon}_n \right)^{-1} \boldsymbol{s}_{n,b}}, \qquad (15)$$

where the failure probability is

$$\delta = 2MB \exp \left\{ -\frac{\nu^2}{2C_R^2} \right\},$$

yielding that $\nu = \sqrt{2C_R^2 \log(2MB/\delta)}$.

**Bounding** $\left| \boldsymbol{s}_{n,b}^{\mathsf{T}} \left( \boldsymbol{I}_d + \boldsymbol{\Upsilon}_n \right)^{-1} \boldsymbol{\theta}_A^* \right|$:

Since $\boldsymbol{\Upsilon}_n$ is positive semi-definite, combining with the Hölder inequality, we obtain

$$
\begin{aligned}
\left| \boldsymbol{s}_{n,b}^{\mathsf{T}} \left( \boldsymbol{I}_d + \boldsymbol{\Upsilon}_n \right)^{-1} \boldsymbol{\theta}_A^* \right| &\leq \left\| \boldsymbol{\theta}_A^* \right\|_2 \left\| \left( \boldsymbol{I}_d + \boldsymbol{\Upsilon}_n \right)^{-1} \boldsymbol{s}_{n,b} \right\|_2 \\
&= \left\| \boldsymbol{\theta}_A^* \right\|_2 \sqrt{ \boldsymbol{s}_{n,b} \left( \boldsymbol{I}_d + \boldsymbol{\Upsilon}_n \right)^{-1} \left( \boldsymbol{I}_d + \boldsymbol{\Upsilon}_n \right)^{-1} \boldsymbol{s}_{n,b}} \\
&\leq \left\| \boldsymbol{\theta}_A^* \right\|_2 \sqrt{ \boldsymbol{s}_{n,b} \left( \boldsymbol{I}_d + \boldsymbol{\Upsilon}_n \right)^{-1} \left( \boldsymbol{I}_d + \boldsymbol{\Upsilon}_n \right) \left( \boldsymbol{I}_d + \boldsymbol{\Upsilon}_n \right)^{-1} \boldsymbol{s}_{n,b}} \\
&= \left\| \boldsymbol{\theta}_A^* \right\|_2 \sqrt{ \boldsymbol{s}_{n,b} \left( \boldsymbol{I}_d + \boldsymbol{\Upsilon}_n \right)^{-1} \boldsymbol{s}_{n,b}}.
\end{aligned}
\qquad (16)
$$

Combing Eq.(15) and Eq.(16) concludes the proof.

Similarly to the proof of Eq.(29), we can obtain that the variance term in full-information setting is smaller than that in partial-information setting. $\qquad \square$

## B    DETAILED PROOFS IN THEORETICAL ANALYSIS

In this section, we provide the instantaneous regret bound in each episode, prove the approximation error of sketching, and analyze the regret for policy updating in the CBB setting. Figure 4 describes the dependence structure of our theoretical results.

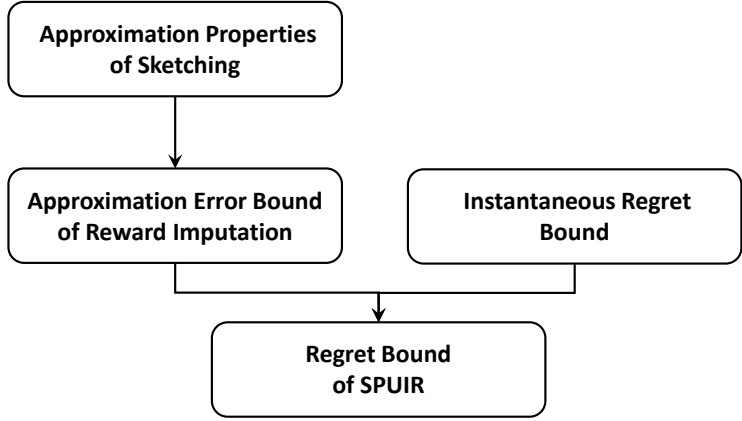

Figure 4: The dependence structure of our theoretical results, where the proof of instantaneous regret bound (Theorem 2) is provided in Appendix B.1, the analysis of approximation properties of sketching (Theorem 6) is given in Appendix B.2, the approximation error bound of reward imputation (Theorem 3) is proven in Appendix B.3, and the regret of SPUIR (Theorem 4) is analyzed in Appendix B.4

### B.1 Proof of Theorem 2

Before we provide the detailed proof of Theorem 2, we first demonstrate a lemma about the convergence and monotonicity of the sum of functions, which is the main tool for analyzing the additional bias of reward imputation.

**Lemma 1** (Convergence and Monotonicity). *Let* $f(n) = \sum_{j=1}^{n} a^{n-j} \cdot g(j)$, *where* $a \in (0, 1)$ *and* $n$ *is a positive integer. Then,*

*1) when* $g(j)$ *is convergent, the limit* $\lim_{n \to \infty} f(n)$ *exists. Moreover,*

$$\lim_{n \to \infty} f(n) = \frac{1}{1-a} \lim_{n \to \infty} g(n). \tag{17}$$

*2)* $f(n)$ *is a monotonic decreasing function if and only if* $g(j)$ *satisfies, for any positive integer* $j \geq 2$,

$$g(j) \leq \begin{cases} (j-1)a^{j-1}g(1) & a = 1/2, \\ \dfrac{(1-a)\left[a^{j-1} - (1-a)^{j-1}\right]}{2a-1}g(1) & a \neq 1/2. \end{cases} \tag{18}$$

*Proof.* Letting $b(j) = a^{-j} \cdot g(j), \forall j \in [n]$, and $S(n) = \sum_{j=1}^{n} b(j)$, $f(n)$ can be rewritten as $f(n) = a^n S(n)$.

1) Rewriting $f(n) = S(n)/a^{-n}$, from the Stolz's theorem, we have

$$\begin{aligned} \lim_{n \to \infty} f(n) &= \lim_{n \to \infty} \frac{S(n) - S(n-1)}{a^{-n} - a^{-(n-1)}} \\ &= \lim_{n \to \infty} \frac{b(n)}{a^{-n} - a^{-(n-1)}} \\ &= \lim_{n \to \infty} \frac{a^{-n} \cdot g(n)}{a^{-n} - a^{-(n-1)}} \\ &= \frac{1}{1-a} \lim_{n \to \infty} g(n). \end{aligned}$$

2) The condition that $f(\cdot)$ is a monotonic decreasing function is equivalent to the following condition: for and positive integer $n$,

$$\begin{aligned} f(n+1) &\leq f(n) \\ \Leftrightarrow a^{n+1}S(n+1) &\leq a^n S(n) \\ \Leftrightarrow a[S(n) + b(n+1)] &\leq S(n) \\ \Leftrightarrow b(n+1) &\leq (1/a - 1)\, S(n). \end{aligned} \tag{19}$$

From the equivalent condition Eq.(19), we obtain the following recursion formula:

$$b(n+1) \leq (1/a - 1)\, S(n)$$

$$b(n) \leq (1/a - 1) \sum_{j=1}^{n-1} b(j)$$

$$\vdots$$

$$b(3) \leq (1/a - 1)\left[b(1) + b(2)\right]$$

$$b(2) \leq (1/a - 1)\, b(1),$$

yielding that, for any positive integer $j \geq 2$,

$$b(j) \leq \left[(1/a - 1) + (1/a - 1)^2 + \cdots + (1/a - 1)^{j-1}\right] b(1). \tag{20}$$

From Eq.(20) , for $a \neq 1/2$,

$$
\begin{aligned}
b(j) &\leq \frac{(1/a - 1)\left[1 - (1/a - 1)^{j-1}\right]}{1 - (1/a - 1)} \, b(1) \\
&= \frac{(1 - a)\left[1 - (1/a - 1)^{j-1}\right]}{2a - 1} \, b(1),
\end{aligned}
\tag{21}
$$

and substituting the definition of $b(j)$ into Eq.(21) yields the equivalent condition

$$
g(j) \leq \frac{(1 - a)\left[1 - (1/a - 1)^{j-1}\right]}{2a - 1} a^{j-1} \, g(1).
$$

For $a = 1/2$, we have the condition $b(j) \leq (j - 1)b(1)$, which is equivalent to

$$
a^{-j} \cdot g(j) \leq (j - 1)a^{-1} \cdot g(1) \iff g(j) \leq (j - 1)a^{j-1}g(1).
$$

$\square$

Next, we provide the detailed proof of Theorem 2.

*Proof of Theorem 2.* From the formulation of $\bar{\boldsymbol{\theta}}_A^n$ and the triangle inequality, we can obtain that, for each action $A \in \mathcal{A}$,

$$
\begin{aligned}
&\left|\langle \bar{\boldsymbol{\theta}}_A^n, \boldsymbol{s}_{n,b} \rangle - \langle \boldsymbol{\theta}_A^*, \boldsymbol{s}_{n,b} \rangle\right| \\
&= \left|\boldsymbol{s}_{n,b}^\mathsf{T} (\boldsymbol{\Psi}_A^n)^{-1} \left(\boldsymbol{b}_A^n + \gamma \hat{\boldsymbol{b}}_A^n\right) - \boldsymbol{s}_{n,b}^\mathsf{T} \boldsymbol{\theta}_A^*\right| \\
&= \left|\boldsymbol{s}_{n,b}^\mathsf{T} (\boldsymbol{\Psi}_A^n)^{-1} \left[\left(\boldsymbol{L}_A^{n-1}\right)^\mathsf{T} \boldsymbol{T}_A^{n-1} + \gamma \left(\hat{\boldsymbol{L}}_A^{n-1}\right)^\mathsf{T} \hat{\boldsymbol{T}}_A^{n-1} - \boldsymbol{\Psi}_A^n \boldsymbol{\theta}_A^*\right]\right| \\
&= \left|\boldsymbol{s}_{n,b}^\mathsf{T} (\boldsymbol{\Psi}_A^n)^{-1} \left[\left(\boldsymbol{L}_A^{n-1}\right)^\mathsf{T} \boldsymbol{T}_A^{n-1} + \gamma \left(\hat{\boldsymbol{L}}_A^{n-1}\right)^\mathsf{T} \hat{\boldsymbol{T}}_A^{n-1} - \left(\lambda \boldsymbol{I}_d + \boldsymbol{\Phi}_A^n + \gamma \hat{\boldsymbol{\Phi}}_A^n\right) \boldsymbol{\theta}_A^*\right]\right| \\
&= \left|\boldsymbol{s}_{n,b}^\mathsf{T} (\boldsymbol{\Psi}_A^n)^{-1} \left\{\left(\boldsymbol{L}_A^{n-1}\right)^\mathsf{T} \boldsymbol{T}_A^{n-1} + \gamma \left(\hat{\boldsymbol{L}}_A^{n-1}\right)^\mathsf{T} \hat{\boldsymbol{T}}_A^{n-1} - \right.\right. \\
&\qquad \left.\left.\left[\lambda \boldsymbol{I}_d + \left(\boldsymbol{L}_A^{n-1}\right)^\mathsf{T} \boldsymbol{L}_A^{n-1} + \gamma \left(\hat{\boldsymbol{L}}_A^{n-1}\right)^\mathsf{T} \hat{\boldsymbol{L}}_A^{n-1}\right] \boldsymbol{\theta}_A^*\right\}\right| \\
&= \left|\boldsymbol{s}_{n,b}^\mathsf{T} (\boldsymbol{\Psi}_A^n)^{-1} \left(\boldsymbol{L}_A^{n-1}\right)^\mathsf{T} \left(\boldsymbol{T}_A^{n-1} - \boldsymbol{L}_A^{n-1} \boldsymbol{\theta}_A^*\right) - \lambda \boldsymbol{s}_{n,b}^\mathsf{T} (\boldsymbol{\Psi}_A^n)^{-1} \boldsymbol{\theta}_A^* + \right. \\
&\qquad \left.\boldsymbol{s}_{n,b}^\mathsf{T} (\boldsymbol{\Psi}_A^n)^{-1} \gamma \left(\hat{\boldsymbol{L}}_A^{n-1}\right)^\mathsf{T} \left(\hat{\boldsymbol{T}}_A^{n-1} - \hat{\boldsymbol{L}}_A^{n-1} \boldsymbol{\theta}_A^*\right)\right| \\
&\leq \underbrace{\left|\boldsymbol{s}_{n,b}^\mathsf{T} (\boldsymbol{\Psi}_A^n)^{-1} \left(\boldsymbol{L}_A^{n-1}\right)^\mathsf{T} \left(\boldsymbol{T}_A^{n-1} - \boldsymbol{L}_A^{n-1} \boldsymbol{\theta}_A^*\right)\right|}_{X_A^{(1)}} + \underbrace{\lambda \left|\boldsymbol{s}_{n,b}^\mathsf{T} (\boldsymbol{\Psi}_A^n)^{-1} \boldsymbol{\theta}_A^*\right|}_{X_A^{(2)}} + \\
&\qquad \underbrace{\left|\boldsymbol{s}_{n,b}^\mathsf{T} (\boldsymbol{\Psi}_A^n)^{-1} \gamma \left(\hat{\boldsymbol{L}}_A^{n-1}\right)^\mathsf{T} \left(\hat{\boldsymbol{T}}_A^{n-1} - \hat{\boldsymbol{L}}_A^{n-1} \boldsymbol{\theta}_A^*\right)\right|}_{X_A^{(3)}}.
\end{aligned}
$$

Next, we bound $X_A^{(1)}$, $X_A^{(2)}$, and $X_A^{(3)}$. For convenience, we drop all the superscripts and subscripts about $n$ and $b$. Similarly to the proof of Theorem 1, we bound $X_A^{(1)} + X_A^{(2)}$ as follows: with probability at least $1 - \delta$,

$$
X_A^{(1)} + X_A^{(2)} \leq (\lambda\|\boldsymbol{\theta}_A^*\|_2 + \nu)\sqrt{\boldsymbol{s}^\mathsf{T} \boldsymbol{\Psi}_A^{-1} \boldsymbol{s}},
\tag{22}
$$

where $\nu = \sqrt{2C_R^2 \log(2MB/\delta)}$. For $X_A^{(3)}$, using the Cauchy-Schwarz inequality, we have

$$
\begin{aligned}
X_A^{(3)} &\leq \gamma \left\|\hat{\boldsymbol{L}}_A \boldsymbol{\Psi}_A^{-1} \boldsymbol{s}\right\|_2 \left\|\hat{\boldsymbol{T}}_A - \hat{\boldsymbol{L}}_A \boldsymbol{\theta}_A^*\right\|_2 \\
&= \sqrt{\gamma}\sqrt{\boldsymbol{s}^\mathsf{T} \boldsymbol{\Psi}_A^{-1} \left(\gamma \hat{\boldsymbol{L}}_A^\mathsf{T} \hat{\boldsymbol{L}}_A\right) \boldsymbol{\Psi}_A^{-1} \boldsymbol{s}} \left\|\hat{\boldsymbol{T}}_A - \hat{\boldsymbol{L}}_A \boldsymbol{\theta}_A^*\right\|_2 \\
&\leq \sqrt{\gamma}\sqrt{\boldsymbol{s}^\mathsf{T} \boldsymbol{\Psi}_A^{-1} \boldsymbol{s}} \left\|\hat{\boldsymbol{T}}_A - \hat{\boldsymbol{L}}_A \boldsymbol{\theta}_A^*\right\|_2.
\end{aligned}
\tag{23}
$$

Now we need to bound the term $\left\| \widehat{\boldsymbol{T}}_A - \widehat{\boldsymbol{L}}_A \boldsymbol{\theta}_A^* \right\|_2$. Since using the discount parameter $\eta \in (0, 1)$ in Eq.(4) is equivalent to multiplying both the imputed contexts and the imputed rewards by the parameter $\sqrt{\eta}$ in each episode, we have, in the $n$-th episode,

$$\left\| \widehat{\boldsymbol{T}}_A^{n-1} - \widehat{\boldsymbol{L}}_A^{n-1} \boldsymbol{\theta}_A^* \right\|_2 = \left\| \boldsymbol{\Delta}_{n-1}^\eta \right\|_2, \tag{24}$$

where $\boldsymbol{\Delta}_{n-1}^\eta = \{\eta^{(n-i-1)/2} \, \mathrm{IR}_{i,b}\}_{i \in [n-1], b \in [B]}$ denotes an exponential-decay vector of the instantaneous regrets, and $\mathrm{IR}_{i,b}$ denotes the instantaneous regret at step $b$ in the $i$-th episode, i.e, $\mathrm{IR}_{i,b} = \left| \langle \bar{\boldsymbol{\theta}}_A^i, \boldsymbol{s}_{i,b} \rangle - \langle \boldsymbol{\theta}_A^*, \boldsymbol{s}_{i,b} \rangle \right|$. From Eq.(24), letting

$$\mathrm{CIR}_i = \sum_{b \in [B]} \mathrm{IR}_{i,b} \tag{25}$$

be the cumulative instantaneous regret in the $i$-th episode, we can obtain the upper bound of Eq.(24) as follows:

$$\begin{aligned}
\left\| \widehat{\boldsymbol{T}}_A^{n-1} - \widehat{\boldsymbol{L}}_A^{n-1} \boldsymbol{\theta}_A^* \right\|_2 &= \left\| \boldsymbol{\Delta}_{n-1}^\eta \right\|_2 \\
&\leq \left\| \boldsymbol{\Delta}_{n-1}^\eta \right\|_1 \\
&= \sum_{i \in [n-1], b \in [B]} \left| \eta^{(n-i-1)/2} \, \mathrm{IR}_{i,b} \right| \\
&= \sum_{i \in [n-1]} \eta^{(n-i-1)/2} \, \mathrm{CIR}_i \\
&= \eta^{-\frac{1}{2}} f_{\mathrm{Imp}}(n),
\end{aligned} \tag{26}$$

where

$$f_{\mathrm{Imp}}(n) := \sum_{i \in [n-1]} (\sqrt{\eta})^{n-i} \, \mathrm{CIR}_i. \tag{27}$$

From monotone bounded theorem, we have that the limit of $\mathrm{CIR}_i$ exists. From Eq.(17) in Lemma 1, we get that $f_{\mathrm{Imp}}(n)$ is convergent and then has an upper bound. We denotes the upper bound of $f_{\mathrm{Imp}}(n)$ by $C_{\mathrm{Imp}} > 0$, and then from Eq.(26) we have

$$\left\| \widehat{\boldsymbol{T}}_A^{n-1} - \widehat{\boldsymbol{L}}_A^{n-1} \boldsymbol{\theta}_A^* \right\|_2 \leq \eta^{-\frac{1}{2}} C_{\mathrm{Imp}}. \tag{28}$$

Substituting Eq.(28) into Eq.(23) yields the upper bound of $X_A^{(3)}$.

Then, we prove that

$$\sqrt{\boldsymbol{s}^\intercal (\boldsymbol{\Psi}_A^n)^{-1} \boldsymbol{s}} \leq \sqrt{\boldsymbol{s}^\intercal (\lambda \boldsymbol{I}_d + \boldsymbol{\Phi}_A^n)^{-1} \boldsymbol{s}}. \tag{29}$$

holds, which is equivalent to

$$\boldsymbol{s}^\intercal \left( \lambda \boldsymbol{I}_d + \boldsymbol{\Phi} + \gamma \widehat{\boldsymbol{\Phi}} \right)^{-1} \boldsymbol{s} \leq \boldsymbol{s}^\intercal (\lambda \boldsymbol{I}_d + \boldsymbol{\Phi})^{-1} \boldsymbol{s}. \tag{30}$$

Letting $\boldsymbol{\Theta} = \lambda \boldsymbol{I}_d + \boldsymbol{\Phi}$, by Sherman-Morrison-Woodbury formula, we have

$$\begin{aligned}
\left( \boldsymbol{\Theta} + \gamma \widehat{\boldsymbol{\Phi}} \right)^{-1} = \left( \boldsymbol{\Theta} + \gamma \widehat{\boldsymbol{S}}^\intercal \widehat{\boldsymbol{S}} \right)^{-1} &= \boldsymbol{\Theta}^{-1} - \gamma \boldsymbol{\Theta}^{-1} \widehat{\boldsymbol{S}}^\intercal \left( \boldsymbol{I}_d + \gamma \widehat{\boldsymbol{S}} \boldsymbol{\Theta}^{-1} \widehat{\boldsymbol{S}}^\intercal \right)^{-1} \widehat{\boldsymbol{S}} \boldsymbol{\Theta}^{-1} \\
&= \boldsymbol{\Theta}^{-1} - \boldsymbol{\Theta}^{-1} \widehat{\boldsymbol{S}}^\intercal \left( \frac{\boldsymbol{I}_d}{\gamma} + \widehat{\boldsymbol{S}} \boldsymbol{\Theta}^{-1} \widehat{\boldsymbol{S}}^\intercal \right)^{-1} \widehat{\boldsymbol{S}} \boldsymbol{\Theta}^{-1},
\end{aligned} \tag{31}$$

yielding that Eq.(30) is equivalent to

$$\boldsymbol{s}^\intercal \boldsymbol{\Gamma} \boldsymbol{s} \geq 0, \tag{32}$$

where

$$\boldsymbol{\Gamma} = \boldsymbol{\Theta}^{-1} \widehat{\boldsymbol{S}}^\intercal \left( \boldsymbol{I}_d/\gamma + \widehat{\boldsymbol{S}} \boldsymbol{\Theta}^{-1} \widehat{\boldsymbol{S}}^\intercal \right)^{-1} \widehat{\boldsymbol{S}} \boldsymbol{\Theta}^{-1}.$$

Let $\boldsymbol{S} = \boldsymbol{U}_d \boldsymbol{\Sigma}_d^{1/2} \boldsymbol{V}_d^\intercal$, $\widehat{\boldsymbol{S}} = \widehat{\boldsymbol{U}}_d \widehat{\boldsymbol{\Sigma}}_d^{1/2} \widehat{\boldsymbol{V}}_d^\intercal$ be the Singular Value Decomposition (SVD) of $\boldsymbol{S}$ and $\widehat{\boldsymbol{S}}$, respectively. Note that $\boldsymbol{\Phi} = \boldsymbol{V}_d \boldsymbol{\Sigma}_d \boldsymbol{V}_d^\intercal$, $\widehat{\boldsymbol{\Phi}} = \widehat{\boldsymbol{V}}_d \widehat{\boldsymbol{\Sigma}}_d \widehat{\boldsymbol{V}}_d^\intercal$. We can obtain that $\boldsymbol{\Gamma}$ is a square symmetric positive semi-definite matrix, since $\boldsymbol{\Gamma}$ can be decomposed into

$$\boldsymbol{\Gamma} = \boldsymbol{Q}^\intercal \boldsymbol{Q},$$

where $P_\gamma \Lambda_\gamma P_\gamma^\mathsf{T}$ is the SVD of $I_d/\gamma + \widehat{S}\Theta^{-1}\widehat{S}^\mathsf{T}$ and

$$Q = \Lambda_\gamma^{-1/2} P_\gamma^\mathsf{T} \widehat{S}\Theta^{-1}.$$

Thus, Eq.(32) holds, yielding that Eq.(30) also holds.

Finally, we prove that a larger imputation rate $\gamma$ leads to a smaller variance term $\sqrt{s^\mathsf{T}(\Psi)^{-1}s}$. From Eq.(31), the variance term can be represented as follows:

$$\sqrt{s^\mathsf{T}(\Psi)^{-1}s} = \left[ s^\mathsf{T}\Theta^{-1}s - s^\mathsf{T}\Theta^{-1}\widehat{S}^\mathsf{T} M_\gamma^{-1} \widehat{S}\Theta^{-1}s \right]^{1/2}, \tag{33}$$

where $M_\gamma = I_d/\gamma + \widehat{S}\Theta^{-1}\widehat{S}^\mathsf{T}$. Letting $M_\gamma = U_{M_\gamma} \Lambda_{M_\gamma} U_{M_\gamma}^\mathsf{T}$ be the SVD of $M_\gamma$, and $z = U_{M_\gamma}^\mathsf{T} \widehat{S}\Theta^{-1}s$, from Eq.(33) we can written the variance term as follows:

$$\sqrt{s^\mathsf{T}(\Psi)^{-1}s} = \left[ s^\mathsf{T}\Theta^{-1}s - z^\mathsf{T}\Lambda_{M_\gamma}^{-1} z \right]^{1/2}. \tag{34}$$

In Eq.(34), we can observed that

$$z^\mathsf{T}\Lambda_{M_\gamma}z = \|(\Lambda_{M_\gamma})^{-1/2}z\|_2^2 \in \left[ \frac{1}{\sigma_{\max}(M) + 1/\gamma}\|z\|_2^2, \ \frac{1}{\sigma_{\min}(M) + 1/\gamma}\|z\|_2^2 \right]$$

where $M = \widehat{S}\Theta^{-1}\widehat{S}^\mathsf{T}$, which indicates that a larger imputation rate $\gamma$ leads to a smaller variance term. $\qquad\square$

Finally, we provide a deeper understanding of the additional bias in Theorem 2.

**Remark 4** (Controllable Bias). *Our reward imputation approach incurs a bias term $\gamma^{\frac{1}{2}}\eta^{-\frac{1}{2}}C_{\mathrm{Imp}}$ in addition to the two bias terms $\lambda\|\theta_A^*\|_2$ and $\nu$ that exist in every UCB-based policy. But this additional bias term is controllable due to the presence of imputation rate $\gamma$ that can help controlling the additional bias. Moreover, from the proof of Eq.(26), we can obtain that, the term $C_{\mathrm{Imp}}$ in the additional bias can be replaced by a function $f_{\mathrm{Imp}}(n)$ (defined in Eq.(27)), and the additional bias term turns out to be $\gamma^{\frac{1}{2}}\eta^{-\frac{1}{2}} f_{\mathrm{Imp}}(n)$. Since $f_{\mathrm{Imp}}(n)$ has the same functional form as the function $f(n)$ in Lemma 1, we can find the conditions that $f_{\mathrm{Imp}}(n)$ is monotonic decreasing following Eq.(18) in Lemma 1. Specifically, letting $\mathrm{CIR}_i$ be the cumulative instantaneous regret in the $i$-th episode defined in Eq.(25),*

*1) when $\sqrt{\eta} \neq 1/2$, the condition of a monotonic decreasing function $f_{\mathrm{Imp}}(\cdot)$ is equivalent to, for any positive integer $i \geq 2$,*

$$\mathrm{CIR}_i \leq \frac{(1-\sqrt{\eta})\left[\sqrt{\eta}^{i-1} - (1-\sqrt{\eta})^{i-1}\right]}{2\sqrt{\eta}-1}\mathrm{CIR}_1,$$

*indicating that the regret after $N$ episodes satisfies*

$$\begin{aligned}
\sum_{2\leq i\leq N}\mathrm{CIR}_i &\leq \mathrm{CIR}_1 \sum_{2\leq i\leq N} \frac{(1-\sqrt{\eta})\left[\sqrt{\eta}^{i-1} - (1-\sqrt{\eta})^{i-1}\right]}{2\sqrt{\eta}-1} \\
&= \mathrm{CIR}_1 \frac{1-\sqrt{\eta}}{2\sqrt{\eta}-1} \sum_{2\leq i\leq N}\left[\sqrt{\eta}^{i-1} - (1-\sqrt{\eta})^{i-1}\right] \\
&= \mathrm{CIR}_1 \frac{1}{\sqrt{\eta}(2\sqrt{\eta}-1)}\left[2\sqrt{\eta}-1 + (1-\sqrt{\eta})^{N+1} - (\sqrt{\eta})^{N+1}\right] \\
&= \mathrm{CIR}_1 \frac{1}{\sqrt{\eta}}\left[1 + \frac{(1-\sqrt{\eta})^{N+1} - (\sqrt{\eta})^{N+1}}{2\sqrt{\eta}-1}\right]. \tag{35}
\end{aligned}$$

*2) for the case $\sqrt{\eta} = 1/2$, the condition of a monotonic decreasing function $f_{\mathrm{Imp}}(\cdot)$ is equivalent to $\mathrm{CIR}_i \leq (i-1)(\sqrt{\eta})^{i-1}\mathrm{CIR}_1$ for any positive integer $i \geq 2$, indicating that the regret after*

*N episodes satisfies*

$$\sum_{2 \leq i \leq N} \mathrm{CIR}_i \leq \mathrm{CIR}_1 \sum_{2 \leq i \leq N} (i-1)(\sqrt{\eta})^{i-1}$$

$$= \frac{\sqrt{\eta}}{(1-\sqrt{\eta})^2}\mathrm{CIR}_1 - \left[\frac{1}{(1-\sqrt{\eta})^2} + \frac{N-1}{1-\sqrt{\eta}}\right](\sqrt{\eta})^N\mathrm{CIR}_1$$

$$= \left(2 - \frac{1+N}{2^{N-1}}\right)\mathrm{CIR}_1$$

$$= \left[\frac{1}{\sqrt{\eta}} - (1+N)\sqrt{\eta}^{N-1}\right]\mathrm{CIR}_1. \tag{36}$$

*From Eq.(35) and Eq.(36), we can conclude that a monotonic decreasing function $f_{\mathrm{Imp}}(\cdot)$ indicates the upper bound of regret after $N$ episodes is of order $O(\mathrm{CIR}_1/\sqrt{\eta})$. The conclusion also indicates that setting the discount parameter as $\sqrt{\eta} = \Theta(\mathrm{CIR}_1/N)$ achieves a $O(N)$ regret bound (i.e., a $\tilde{O}(\sqrt{dT})$ regret bound following Remark 3). Note that setting the discount parameter as $\sqrt{\eta} = \Theta(\mathrm{CIR}_1/N)$ is a mild condition, since the cumulative instantaneous regret $\mathrm{CIR}_1$ is typically of order $O(B)$ ($B = O(\sqrt{T/d})$ in Remark 3) yielding that $\sqrt{\eta} = \Theta(d^{-1})$. Overall, since a larger imputation rate $\gamma$ leads to a smaller variance while increasing the bias (variance analysis can be found in Remark 2), $\gamma$ controls a trade-off between the bias term and the variance term. When $f_{\mathrm{Imp}}$ is a monotonic decreasing function w.r.t. number of episodes $n$, the additional bias term $\gamma^{\frac{1}{2}}\eta^{-\frac{1}{2}}f_{\mathrm{Imp}}(n)$ can be easily controlled, e.g., gradually increasing $\gamma$ with the number of episodes, avoiding the large bias from $f_{\mathrm{Imp}}(n)$ at the beginning of reward imputation. We design a rate-scheduled approach for choosing the imputation rate $\gamma$ in Section 5.*

**Remark 5** (Relationship to Exploration and Exploitation Trade-off). *Exploration-exploitation dilemma is the key challenge in online learning under bandit settings. In the full-information setting, agent (e.g., UCB policy) receives the rewards from all the actions, does not need to consider the choice of exploring the feedback mechanisms, and achieves a lower variance part in the regret upper bound (Theorem 1). Along this line, our reward computation approach is proposed to approximate the setting of full-information feedback, which somewhat relaxes the explore/exploit dilemma and also brings a lower variance part and a controllable additional bias part in the regret. Extra information that pushes the policy towards exploitation and away from exploration comes from the estimated reward structures of the non-executed actions maintained in each episode, and the proposed reward imputation can be seen as an effective and efficient tool to capture this extra information.*

### B.2 Approximation Properties of Sketching

Although some error bounds of approximation using SJLT have been proposed (Nelson & Nguyên, 2013; Kane & Nelson, 2014; Bourgain et al., 2015), it is still unknown what is the lower bound of the sketch size while applying SJLT to the sketched ridge regression problem in our SPUIR. To address this issue, we first prove two approximation properties of SJLT which are necessary to achieve approximation error bound of the sketched ridge regression using SJLT. For convenience, we drop all the superscripts and subscripts in these theoretical results.

**Lemma 2** ((Nelson & Nguyên, 2013)). *Let $U \in \mathbb{R}^{L \times d}$ be a matrix with orthonormal columns, $\Pi \in \mathbb{R}^{c \times L}$ the SJLT. Assuming that $D = \Theta(\varepsilon_\sigma^{-1}\log^3(d\delta_0^{-1}))$ for $\Pi$, $\varepsilon_\sigma \in (0,1)$ and $d \leq c$, with probability at least $1 - \delta_0$ all singular values of $\Pi U$*

$$\sigma_i(\Pi U) = 1 \pm \varepsilon_\sigma, \quad i \in [d],$$

*as long as*

$$c \geq \frac{d\log^8\left(d\delta_0^{-1}\right)}{\varepsilon_\sigma^2}.$$

*Further, this holds if the hash function $h$ and $\sigma$ defining the $\Pi$ is $\Omega\left(\log(d\delta_0^{-1})\right)$-wise independent.*

**Theorem 6** (Approximation Properties of SJLT). *Let $U \in \mathbb{R}^{L \times d}$ be a matrix with orthonormal columns, and $A$ be a matrix of any proper size. If $\Pi \in \mathbb{R}^{c \times L}$ is the SJLT satisfying the assumptions in Lemma 2, and $d \leq c \leq L$, then $\Pi$ has the following two properties:*

1) *Subspace embedding property: set* $c = \Omega\left(d \operatorname{polylog}\left(d\delta_{\mathrm{s}}^{-1}\right)/\varepsilon_{\mathrm{s}}^2\right)$, *for* $\varepsilon_{\mathrm{s}} \in (0,1)$, *with probability at least* $1 - \delta_{\mathrm{s}}$,

$$\left\|U^{\mathsf{T}}\Pi^{\mathsf{T}}\Pi U - I_d\right\|_2 \leq \varepsilon_{\mathrm{s}};$$

2) *Matrix multiplication property: set* $c = \Omega(d/(\varepsilon_{\mathrm{m}}\delta_{\mathrm{m}}))$, *for* $\varepsilon_{\mathrm{m}} \in (0,1)$, *with probability at least* $1 - \delta_{\mathrm{m}}$,

$$\left\|U^{\mathsf{T}}\Pi^{\mathsf{T}}\Pi A - U^{\mathsf{T}}A\right\|_{\mathrm{F}}^2 \leq \varepsilon_{\mathrm{m}}\|A\|_{\mathrm{F}}^2.$$

**Proof of Theorem 6.** 1) From Lemma 2, by setting $c = \Omega\left(d \operatorname{polylog}\left(d\delta_{\mathrm{s}}^{-1}\right)/\varepsilon_0^2\right)$, we can obtain the upper bounds of eigenvalues: with probability at least $1 - \delta_{\mathrm{s}}$,

$$\lambda_i\left(U^{\mathsf{T}}\Pi^{\mathsf{T}}\Pi U\right) = \sigma_i^2(\Pi U) \in [(1-\varepsilon_\sigma)^2, (1+\varepsilon_\sigma)^2] \subseteq [1-2\varepsilon_\sigma, 1+3\varepsilon_\sigma], \qquad (37)$$

which yields that

$$\left|\lambda_i\left(U^{\mathsf{T}}\Pi^{\mathsf{T}}\Pi U - I_d\right)\right| \leq 3\varepsilon_\sigma. \qquad (38)$$

Eq.(39) is equivalent to

$$\left\|U^{\mathsf{T}}\Pi^{\mathsf{T}}\Pi U - I_d\right\|_2 \leq 3\varepsilon_\sigma.$$

Letting $\varepsilon_{\mathrm{s}} = 3\varepsilon_\sigma$ and $\varepsilon_\sigma \in (0, 1/3)$ yields the subspace embedding property.

2) From Lemma 1 in (Zhang & Liao, 2019), we have

$$\mathbb{E}\left[\left\|U^{\mathsf{T}}\Pi^{\mathsf{T}}\Pi A - U^{\mathsf{T}}A\right\|_{\mathrm{F}}^2\right] \leq \frac{2}{c}\|U\|_{\mathrm{F}}^2\|A\|_{\mathrm{F}}^2 = \frac{2d}{c}\|A\|_{\mathrm{F}}^2. \qquad (39)$$

Combining Eq.(39) with the Markov's inequality, we obtain that, with probability at least $1 - \delta_{\mathrm{m}}$,

$$\left\|U^{\mathsf{T}}\Pi^{\mathsf{T}}\Pi A - U^{\mathsf{T}}A\right\|_{\mathrm{F}}^2 \leq \frac{2d}{\delta_{\mathrm{m}}c}\|A\|_{\mathrm{F}}^2.$$

Letting $\varepsilon_{\mathrm{m}} = \dfrac{2d}{\delta_{\mathrm{m}}c}$ yields the matrix multiplication property.

$\square$

### B.3 Proof of Theorem 3

Next, using the approximation properties of SJLT in Theorem 6, we prove that the objective function value of the imputation regularized ridge regression problem for reward imputation can be approximated well with a relative-error bound. Moreover, we prove that the solution solving the sketched ridge regression problem for reward imputation is also a good approximation of the solution solving the imputation regularized ridge regression. The following theorem is a detailed version of Theorem 3.

**Theorem 7** (Approximation Error Bound of Imputation using Sketching, Detailed Version of Theorem 3). *Let* $\gamma \in [0,1]$ *be the imputation rate,* $\lambda > 0$ *the regularization parameter,* $\Pi \in \mathbb{R}^{c \times L}$ *and* $\widehat{\Pi} \in \mathbb{R}^{c \times \widehat{L}}$ *be the SJLT, and* $L \in \mathbb{R}^{L \times d}, \widehat{L} \in \mathbb{R}^{\widehat{L} \times d}, T \in \mathbb{R}^L, \widehat{T} \in \mathbb{R}^{\widehat{L}}, \theta \in \mathbb{R}^d$. *Denote the imputation regularized ridge regression function* $F$ *and sketched ridge regression function* $F^{\mathrm{S}}$ *for reward imputation by*

$$F(\theta) = \|L\theta - T\|_2^2 + \gamma\left\|\widehat{L}\theta - \widehat{T}\right\|_2^2 + \lambda\|\theta\|_2^2,$$

$$F^{\mathrm{S}}(\theta) = \|\Pi(L\theta - T)\|_2^2 + \gamma\left\|\widehat{\Pi}\left(\widehat{L}\theta - \widehat{T}\right)\right\|_2^2 + \lambda\|\theta\|_2^2,$$

*and the solutions of these regression problems by*

$$\bar{\theta} = \underset{\theta \in \mathbb{R}^d}{\arg\min}\, F(\theta) \quad \text{and} \quad \tilde{\theta} = \underset{\theta \in \mathbb{R}^d}{\arg\min}\, F^{\mathrm{S}}(\theta).$$

*Let* $\delta \in (0, 0.1]$, $\varepsilon \in (0,1)$, $\rho_\lambda = \|L_{\mathrm{all}}\|_2^2/(\|L_{\mathrm{all}}\|_2^2 + \lambda)$. *For* $\Pi$ *and* $\widehat{\Pi}$, *assuming that* $D = \Theta(\varepsilon^{-1}\log^3(d\delta^{-1}))$ *and*

$$c = \Omega\left(d \operatorname{polylog}\left(d\delta^{-1}\right)/\varepsilon^2\right),$$

*with probability at least $1 - \delta$,*

$$F(\tilde{\boldsymbol{\theta}}) \leq (1 + \rho_\lambda \varepsilon) F(\bar{\boldsymbol{\theta}}), \tag{40}$$

$$\|\tilde{\boldsymbol{\theta}} - \bar{\boldsymbol{\theta}}\|_2 \leq \frac{\sqrt{\rho_\lambda \varepsilon F(\bar{\boldsymbol{\theta}})}}{\sigma_{\min}(\boldsymbol{L}_{\text{all}}^\lambda)}, \tag{41}$$

*where $\boldsymbol{L}_{\text{all}}^\lambda = \left[\boldsymbol{L}; \sqrt{\gamma}\widehat{\boldsymbol{L}}; \sqrt{\lambda}\boldsymbol{I}_d\right] \in \mathbb{R}^{(L+\widehat{L}+d)\times d}$. Furthermore, if there is a constant fraction of the norm of $\boldsymbol{T}_{\text{all}}^0$ lies in the column space of $L_{\text{all}}^\lambda$, then Eq.(41) can be strengthened. Formally, assuming that a mild structural assumption on the context matrix and the reward vector is satisfied, i.e., $\|\boldsymbol{U}_{\text{all}}\boldsymbol{U}_{\text{all}}^\intercal \boldsymbol{T}_{\text{all}}^0\|_2 \geq \xi \|\boldsymbol{T}_{\text{all}}^0\|_2$ with a constant $\xi \in (0, 1]$, then with probability at least $1 - \delta$,*

$$\|\tilde{\boldsymbol{\theta}} - \bar{\boldsymbol{\theta}}\|_2 \leq \left(\kappa(\boldsymbol{L}_{\text{all}}^\lambda)\sqrt{\xi^{-2} - 1}\right)\sqrt{\rho_\lambda \epsilon}\|\bar{\boldsymbol{\theta}}\|_2, \tag{42}$$

*where $\kappa(\boldsymbol{A})$ denotes the condition number of $\boldsymbol{A}$, $\boldsymbol{T}_{\text{all}}^0 = [\boldsymbol{T}; \widehat{\boldsymbol{T}}; \boldsymbol{0}_d] \in \mathbb{R}^{(L+\widehat{L}+d)}$, and $\boldsymbol{L}_{\text{all}}^\lambda = \boldsymbol{U}_{\text{all}}\boldsymbol{\Sigma}_{\text{all}}\boldsymbol{V}_{\text{all}}^\intercal$ is the SVD of $\boldsymbol{L}_{\text{all}}^\lambda$.*

**Proof of Theorem 7.** We first introduce some more notation of block matrices that will simplify the proof of the theorem:

$$\boldsymbol{\Pi}_{\text{all}} = \begin{pmatrix} \boldsymbol{\Pi} & \boldsymbol{O} \\ \boldsymbol{O} & \widehat{\boldsymbol{\Pi}} \end{pmatrix}, \quad \boldsymbol{L}_{\text{all}} = \begin{pmatrix} \boldsymbol{L} \\ \sqrt{\gamma}\widehat{\boldsymbol{L}} \end{pmatrix}, \quad \boldsymbol{T}_{\text{all}} = \begin{pmatrix} \boldsymbol{T} \\ \widehat{\boldsymbol{T}} \end{pmatrix}. \tag{43}$$

Then the regression functions can be rewritten as follows:

$$F(\boldsymbol{\theta}) = \|\boldsymbol{L}_{\text{all}}\boldsymbol{\theta} - \boldsymbol{T}_{\text{all}}\|_2^2 + \lambda\|\boldsymbol{\theta}\|_2^2, \qquad F^{\text{S}}(\boldsymbol{\theta}) = \|\boldsymbol{\Pi}_{\text{all}}(\boldsymbol{L}_{\text{all}}\boldsymbol{\theta} - \boldsymbol{T}_{\text{all}})\|_2^2 + \lambda\|\boldsymbol{\theta}\|_2^2.$$

Obviously, $\boldsymbol{\Pi}_{\text{all}}$ is still an SJLT. Combining Theorem 6 with theorem 19 in (Wang et al., 2017), we can obtain, setting

$$c = \Omega\left(\max\{d \operatorname{polylog}\left(d\delta_{\text{s}}^{-1}\right)/\varepsilon_{\text{s}}^2, \ d/(\varepsilon_{\text{m}}\delta_{\text{m}})\}\right),$$

with probability at least $1 - (\delta_{\text{s}} + \delta_{\text{m}})$,

$$F(\tilde{\boldsymbol{\theta}}) - F(\bar{\boldsymbol{\theta}}) \leq \rho_\lambda \tau F(\bar{\boldsymbol{\theta}}), \tag{44}$$

where $\rho_\lambda = \frac{\|\boldsymbol{L}_{\text{all}}\|_2^2}{\|\boldsymbol{L}_{\text{all}}\|_2^2 + \lambda}$ and $\tau = \frac{2\max\{\varepsilon_{\text{s}}^2, \varepsilon_{\text{m}}\}}{1 - \varepsilon_{\text{s}}}$. Letting $\varepsilon_{\text{s}} = \varepsilon_{\text{m}} := \varepsilon_0$, Eq.(44) can be rewritten as

$$F(\tilde{\boldsymbol{\theta}}) - F(\bar{\boldsymbol{\theta}}) \leq \frac{2\rho_\lambda \varepsilon_0}{1 - \varepsilon_0} F(\bar{\boldsymbol{\theta}}), \tag{45}$$

Assuming that $\delta_{\text{s}} = \delta_{\text{m}} := \delta/2 \in (0, 0.1]$ and $\varepsilon_0 \in (0, 1/3)$, setting $\epsilon = \frac{2\varepsilon_0}{1 - \varepsilon_0} \in (0, 1)$, from Eq.(45) we obtain the upper bound Eq.(40).

Next, we bound the difference between the solutions solving the sketched ridge regression problem and the original regression problem. Since $\sigma_{\min}^2(\boldsymbol{A})\|\boldsymbol{x}\|_2^2 \leq \|\boldsymbol{A}\boldsymbol{x}\|_2^2$ for any $\boldsymbol{A}$ and $\boldsymbol{x}$ with proper sizes, we have

$$\sigma_{\min}^2(\boldsymbol{L}_{\text{all}})\|\tilde{\boldsymbol{\theta}} - \bar{\boldsymbol{\theta}}\|_2^2 \leq \left\|\boldsymbol{L}_{\text{all}}(\tilde{\boldsymbol{\theta}} - \bar{\boldsymbol{\theta}})\right\|_2^2. \tag{46}$$

The key ingredient of bounding $\|\tilde{\boldsymbol{\theta}} - \bar{\boldsymbol{\theta}}\|_2$ is to bound $\|\boldsymbol{L}_{\text{all}}(\tilde{\boldsymbol{\theta}} - \bar{\boldsymbol{\theta}})\|_2$. Let $\boldsymbol{L}_{\text{all}}^\lambda = \left[\boldsymbol{L}; \sqrt{\gamma}\widehat{\boldsymbol{L}}; \sqrt{\lambda}\boldsymbol{I}_d\right] \in \mathbb{R}^{(L+\widehat{L}+d)\times d}$, $\boldsymbol{T}_{\text{all}}^0 = [\boldsymbol{T}; \widehat{\boldsymbol{T}}; \boldsymbol{0}_d] \in \mathbb{R}^{(L+\widehat{L}+d)}$, $\boldsymbol{L}_{\text{all}}^\lambda = \boldsymbol{U}_{\text{all}}\boldsymbol{\Sigma}_{\text{all}}\boldsymbol{V}_{\text{all}}^\intercal$ be the SVD of $\boldsymbol{L}_{\text{all}}^\lambda$, and denote a matrix with orthonormal columns by $\boldsymbol{U}_{\text{all}}^\perp \in \mathbb{R}^{(L+\widehat{L}+d)\times(L+\widehat{L})}$ which satisfies

$$\boldsymbol{U}_{\text{all}}\boldsymbol{U}_{\text{all}}^\intercal + \boldsymbol{U}_{\text{all}}^\perp(\boldsymbol{U}_{\text{all}}^\perp)^\intercal = \boldsymbol{I}_{L+\widehat{L}+d} \quad \text{and} \quad \boldsymbol{U}_{\text{all}}^\intercal \boldsymbol{U}_{\text{all}}^\perp = \boldsymbol{O}.$$

Then, we can rewrite the solution $\bar{\boldsymbol{\theta}}$ as follows:

$$\bar{\boldsymbol{\theta}} = \operatorname*{arg\,min}_{\boldsymbol{\theta} \in \mathbb{R}^d} F(\boldsymbol{\theta}) = \operatorname*{arg\,min}_{\boldsymbol{\theta} \in \mathbb{R}^d} \left\|\boldsymbol{L}_{\text{all}}^\lambda \boldsymbol{\theta} - \boldsymbol{T}_{\text{all}}^0\right\|_2^2$$

$$= (\boldsymbol{L}_{\text{all}}^\lambda)^\dagger \boldsymbol{T}_{\text{all}}^0 = \boldsymbol{V}_{\text{all}}\boldsymbol{\Sigma}_{\text{all}}^{-1}\boldsymbol{U}_{\text{all}}^\intercal \boldsymbol{T}_{\text{all}}^0,$$

which yields that

$$
\begin{aligned}
\boldsymbol{T}_{\mathrm{all}}^0 - \boldsymbol{L}_{\mathrm{all}}^\lambda \bar{\boldsymbol{\theta}} &= \boldsymbol{T}_{\mathrm{all}}^0 - \boldsymbol{L}_{\mathrm{all}}^\lambda \boldsymbol{V}_{\mathrm{all}} \boldsymbol{\Sigma}_{\mathrm{all}}^{-1} \boldsymbol{U}_{\mathrm{all}}^\intercal \boldsymbol{T}_{\mathrm{all}}^0 \\
&= \boldsymbol{T}_{\mathrm{all}}^0 - \boldsymbol{U}_{\mathrm{all}} \boldsymbol{\Sigma}_{\mathrm{all}} \boldsymbol{V}_{\mathrm{all}}^\intercal \boldsymbol{V}_{\mathrm{all}} \boldsymbol{\Sigma}_{\mathrm{all}}^{-1} \boldsymbol{U}_{\mathrm{all}}^\intercal \boldsymbol{T}_{\mathrm{all}}^0 \\
&= \boldsymbol{T}_{\mathrm{all}}^0 - \boldsymbol{U}_{\mathrm{all}} \boldsymbol{U}_{\mathrm{all}}^\intercal \boldsymbol{T}_{\mathrm{all}}^0 \\
&= \boldsymbol{U}_{\mathrm{all}}^\perp (\boldsymbol{U}_{\mathrm{all}}^\perp)^\intercal \boldsymbol{T}_{\mathrm{all}}^0 .
\end{aligned}
\tag{47}
$$

Thus, $\boldsymbol{T}_{\mathrm{all}}^0 - \boldsymbol{L}_{\mathrm{all}}^\lambda \bar{\boldsymbol{\theta}}$ is orthogonal to $\boldsymbol{U}_{\mathrm{all}}$, and consequently to $\boldsymbol{L}_{\mathrm{all}}^\lambda (\tilde{\boldsymbol{\theta}} - \bar{\boldsymbol{\theta}})$, and we can obtain the following equality by Pythagoras's theorem:

$$
\left\| \boldsymbol{L}_{\mathrm{all}}^\lambda (\tilde{\boldsymbol{\theta}} - \bar{\boldsymbol{\theta}}) \right\|_2^2 = \left\| \boldsymbol{L}_{\mathrm{all}}^\lambda \tilde{\boldsymbol{\theta}} - \boldsymbol{T}_{\mathrm{all}}^0 \right\|_2^2 - \left\| \boldsymbol{L}_{\mathrm{all}}^\lambda \bar{\boldsymbol{\theta}} - \boldsymbol{T}_{\mathrm{all}}^0 \right\|_2^2 .
\tag{48}
$$

Combining Eq.(48) with Eq.(40) yields that

$$
\left\| \boldsymbol{L}_{\mathrm{all}}^\lambda (\tilde{\boldsymbol{\theta}} - \bar{\boldsymbol{\theta}}) \right\|_2^2 = F(\tilde{\boldsymbol{\theta}}) - F(\bar{\boldsymbol{\theta}}) \le \rho_\lambda \varepsilon F(\bar{\boldsymbol{\theta}}) .
\tag{49}
$$

Substituting Eq.(49) into Eq.(46) concludes the proof of Eq.(41).

If we make a mild structural assumption on the context matrix and the reward vector, we can provide a stronger bound of $\|\tilde{\boldsymbol{\theta}} - \bar{\boldsymbol{\theta}}\|_2$. Specifically, assuming that $\|\boldsymbol{U}_{\mathrm{all}} \boldsymbol{U}_{\mathrm{all}}^\intercal \boldsymbol{T}_{\mathrm{all}}^0\|_2 \ge \xi \|\boldsymbol{T}_{\mathrm{all}}^0\|_2$ with a constant $\xi \in (0, 1]$, from Eq.(47) and Pythagoras's theorem we have

$$
\begin{aligned}
F(\bar{\boldsymbol{\theta}}) &= \|\boldsymbol{L}_{\mathrm{all}}^\lambda \bar{\boldsymbol{\theta}} - \boldsymbol{T}_{\mathrm{all}}^0\|_2^2 \\
&= \|\boldsymbol{T}_{\mathrm{all}}^0\|_2^2 - \|\boldsymbol{U}_{\mathrm{all}} \boldsymbol{U}_{\mathrm{all}}^\intercal \boldsymbol{T}_{\mathrm{all}}^0\|_2^2 \\
&\le (\xi^{-2} - 1) \|\boldsymbol{U}_{\mathrm{all}} \boldsymbol{U}_{\mathrm{all}}^\intercal \boldsymbol{T}_{\mathrm{all}}^0\|_2^2 \\
&= (\xi^{-2} - 1) \|\boldsymbol{L}_{\mathrm{all}}^\lambda \bar{\boldsymbol{\theta}}\|_2^2 \\
&\le (\xi^{-2} - 1) \|\boldsymbol{L}_{\mathrm{all}}^\lambda\|_2^2 \, \|\bar{\boldsymbol{\theta}}\|_2^2 \\
&\le (\xi^{-2} - 1) \sigma_{\max}^2(\boldsymbol{L}_{\mathrm{all}}^\lambda) \|\bar{\boldsymbol{\theta}}\|_2^2 .
\end{aligned}
\tag{50}
$$

Combining Eq.(50) with Eq.(41) yields Eq.(42). $\qquad\square$

### B.4 PROOF OF THEOREM 4

*Proof of Theorem 4.* In our sketched policy, letting $C_{\boldsymbol{\theta}^*}^{\max} = \max_{A \in \mathcal{A}} \|\boldsymbol{\theta}_A^*\|_2$, $C_{\mathrm{Imp}} > 0$, $\nu = \sqrt{2 C_R^2 \log(2MB/\delta)}$, and

$$
\omega = \lambda C_{\boldsymbol{\theta}^*}^{\max} + \nu + \gamma^{\frac{1}{2}} \eta^{-\frac{1}{2}} C_{\mathrm{Imp}},
$$

from Eq.(9) in Theorem 2 we obtain that

$$
\left| \langle \bar{\boldsymbol{\theta}}_A^n, \boldsymbol{s}_{n,b} \rangle - \langle \boldsymbol{\theta}_A^*, \boldsymbol{s}_{n,b} \rangle \right| \le \omega \sqrt{\boldsymbol{s}_{n,b}^\intercal \left( \boldsymbol{\Psi}_A^n \right)^{-1} \boldsymbol{s}_{n,b}} .
\tag{51}
$$

Before proving the upper bound of $\left| \left\langle \tilde{\boldsymbol{\theta}}_A^n, \boldsymbol{s}_{n,b} \right\rangle - \langle \boldsymbol{\theta}_A^*, \boldsymbol{s}_{n,b} \rangle \right|$, we need to provide a technical tool as follows. For convenience, we also drop all the superscripts and subscripts. The goal is to find a constant $C_\alpha$ such that

$$
\sqrt{\boldsymbol{s}^\intercal \boldsymbol{\Psi}^{-1} \boldsymbol{s}} \le C_\alpha \sqrt{\boldsymbol{s}^\intercal \boldsymbol{W}^{-1} \boldsymbol{s}},
\tag{52}
$$

which is equivalent to the condition that the matrix $C_\alpha^2 \boldsymbol{W}^{-1} - \boldsymbol{\Psi}^{-1}$ is positive semidefinite. Let $\boldsymbol{L}_{\mathrm{all}}$ and $\boldsymbol{\Pi}_{\mathrm{all}}$ be the matrices defined in Eq.(43), $\boldsymbol{L}_{\mathrm{all}} = \widetilde{\boldsymbol{U}}_{\mathrm{all}} \widetilde{\boldsymbol{\Sigma}}_{\mathrm{all}} \widetilde{\boldsymbol{V}}_{\mathrm{all}}^\intercal$ be the SVD of $\boldsymbol{L}_{\mathrm{all}}$, and $\tilde{\sigma}_1 \ge \tilde{\sigma}_2 \cdots \ge \tilde{\sigma}_d$ be the singular values of $\boldsymbol{L}_{\mathrm{all}}$. Then the $i$-th eigenvalue of $\boldsymbol{\Psi}^{-1} = (\lambda \boldsymbol{I}_d + \boldsymbol{L}_{\mathrm{all}}^\intercal \boldsymbol{L}_{\mathrm{all}})^{-1}$ can be represented as $\lambda_i(\boldsymbol{\Psi}^{-1}) = 1/(\tilde{\sigma}_i^2 + \lambda)$, and the $i$-th eigenvalue of $\boldsymbol{W}^{-1} = (\lambda \boldsymbol{I}_d + \boldsymbol{L}_{\mathrm{all}}^\intercal \boldsymbol{\Pi}_{\mathrm{all}}^\intercal \boldsymbol{\Pi}_{\mathrm{all}} \boldsymbol{L}_{\mathrm{all}})^{-1}$ is $\lambda_i(\boldsymbol{W}^{-1}) = 1/(\hat{\lambda}_i + \lambda)$, where $\hat{\lambda}_i$ is the $i$-th eigenvalue of $\widetilde{\boldsymbol{\Sigma}}_{\mathrm{all}} \widetilde{\boldsymbol{U}}_{\mathrm{all}}^\intercal \boldsymbol{\Pi}_{\mathrm{all}}^\intercal \boldsymbol{\Pi}_{\mathrm{all}} \widetilde{\boldsymbol{U}}_{\mathrm{all}} \widetilde{\boldsymbol{\Sigma}}_{\mathrm{all}}$.

From the Lidskii's theorem and Eq.(37), we have

$$
\hat{\lambda}_i \in [\tilde{\sigma}_d^2 (1 - 2\varepsilon_\sigma), \tilde{\sigma}_1^2 (1 + 3\varepsilon_\sigma)].
\tag{53}
$$

Assuming that the positive semi-definiteness of $C_\alpha^2 \boldsymbol{W}^{-1} - \boldsymbol{\Psi}^{-1}$ is satisfied, we obtain that $C_\alpha^2 \lambda_i(\boldsymbol{W}^{-1}) - \lambda_i(\boldsymbol{\Psi}^{-1}) \geq 0$ for $i \in [d]$, and combining this inequality with Eq.(53) yields that

$$C_\alpha = \sqrt{[\tilde{\sigma}_1^2(1 + 3\varepsilon_\sigma) + \lambda]/(\tilde{\sigma}_d^2 + \lambda)}.$$

From the proof of Theorem 6 and Theorem 3, we can obtain that $\varepsilon_\sigma = \varepsilon/(6 + 3\varepsilon)$, yielding that

$$C_\alpha = \sqrt{\frac{\tilde{\sigma}_1^2[1 + \varepsilon/(2 + \varepsilon)] + \lambda}{\tilde{\sigma}_d^2 + \lambda}},$$

which decreases with increase of $1/\varepsilon$. Similarly to the proof of $C_\alpha$ satisfying Eq.(52), we can obtain that

$$\sqrt{\boldsymbol{s}^\mathsf{T} \boldsymbol{W}^{-1} \boldsymbol{s}} \leq C_{\text{reg}} \sqrt{\boldsymbol{s}^\mathsf{T} \boldsymbol{\Psi}^{-1} \boldsymbol{s}}, \tag{54}$$

provided that

$$C_{\text{reg}} = \sqrt{\frac{\tilde{\sigma}_1^2 + \lambda}{\tilde{\sigma}_d^2[1 - 2\varepsilon/(6 + 3\varepsilon)] + \lambda}}.$$

Obviously, $C_{\text{reg}}$ also decreases with increase of $1/\varepsilon$.

Then, letting $\alpha = \omega C_\alpha$, from Eq.(51) and Eq.(52) we have

$$\begin{aligned}
\left| \left\langle \tilde{\boldsymbol{\theta}}_A^n, \boldsymbol{s}_{n,b} \right\rangle - \left\langle \boldsymbol{\theta}_A^*, \boldsymbol{s}_{n,b} \right\rangle \right| &\leq \left| \left\langle \tilde{\boldsymbol{\theta}}_A^n, \boldsymbol{s}_{n,b} \right\rangle - \left\langle \bar{\boldsymbol{\theta}}_A^n, \boldsymbol{s}_{n,b} \right\rangle \right| + \left| \left\langle \bar{\boldsymbol{\theta}}_A^n, \boldsymbol{s}_{n,b} \right\rangle - \left\langle \boldsymbol{\theta}_A^*, \boldsymbol{s}_{n,b} \right\rangle \right| \\
&\leq \left| \left\langle \tilde{\boldsymbol{\theta}}_A^n - \bar{\boldsymbol{\theta}}_A^n, \boldsymbol{s}_{n,b} \right\rangle \right| + \omega \sqrt{\boldsymbol{s}_{n,b}^\mathsf{T} (\boldsymbol{\Psi}_A^n)^{-1} \boldsymbol{s}_{n,b}} \\
&\leq Y_{n,b} + \alpha \sqrt{\boldsymbol{s}_{n,b}^\mathsf{T} (\boldsymbol{W}_A^n)^{-1} \boldsymbol{s}_{n,b}},
\end{aligned} \tag{55}$$

where $Y_{n,b}$ denotes the upper bound of $\left| \left\langle \tilde{\boldsymbol{\theta}}_A^n - \bar{\boldsymbol{\theta}}_A^n, \boldsymbol{s}_{n,b} \right\rangle \right|$ for any $A \in \mathcal{A}$.

Next, using the compatibility of norm, we give a specific representation of the sum of $Y_{n,b}$ as follows:

$$\begin{aligned}
\sum_{b \in [B]} Y_{n,b} &= \max_{A \in \mathcal{A}} \| \boldsymbol{S}_A^n (\tilde{\boldsymbol{\theta}}_A^n - \bar{\boldsymbol{\theta}}_A^n) \|_1 \\
&\leq \max_{A \in \mathcal{A}} \| \boldsymbol{S}_A^n \|_1 \| \tilde{\boldsymbol{\theta}}_A^n - \bar{\boldsymbol{\theta}}_A^n \|_1 \\
&\leq \max_{A \in \mathcal{A}} \| \boldsymbol{S}_A^n \|_1 \sqrt{d} \| \tilde{\boldsymbol{\theta}}_A^n - \bar{\boldsymbol{\theta}}_A^n \|_2.
\end{aligned} \tag{56}$$

Further, we give a more specific upper bound in Eq.(56) under mild structural assumption in Theorem 3. Let $\kappa_{\text{all}}^{\max}$ denote the maximum of the condition numbers of $\boldsymbol{L}_{\text{all}}^\lambda(A, n)$ for $A \in \mathcal{A}, n \in [N]$, and $\boldsymbol{L}_{\text{all}}^\lambda(A, n) = \left[ \boldsymbol{L}_A^n; \sqrt{\gamma} \widehat{\boldsymbol{L}}_A^n; \sqrt{\lambda} \boldsymbol{I}_d \right]$, and $\boldsymbol{U}_{\text{all}}(A, n)$ be the left singular matrix of $\boldsymbol{L}_{\text{all}}^\lambda(A, n)$. Letting $\boldsymbol{T}_{\text{all}}^0(A, n) = [\boldsymbol{T}_A^n; \widehat{\boldsymbol{T}}_A^n; \boldsymbol{0}_d]$, assuming that $\| \boldsymbol{U}_{\text{all}}(A, n) \boldsymbol{U}_{\text{all}}(A, n)^\mathsf{T} \boldsymbol{T}_{\text{all}}^0(A, n) \|_2 \geq \xi \| \boldsymbol{T}_{\text{all}}^0(A, n) \|_2$ with a constant $\xi \in (0, 1]$, substituting the upper bound Eq.(42) in Theorem 3 into Eq.(56) yields that

$$\sum_{b \in [B]} Y_{n,b} \leq \left( \kappa_{\text{all}}^{\max} \sqrt{\xi^{-2} - 1} \right) C_{\boldsymbol{S}} C_{\bar{\boldsymbol{\theta}}}^{\max} \sqrt{\rho_\lambda \epsilon d}, \tag{57}$$

where $C_{\boldsymbol{S}} = \max_{n \in [N], A \in \mathcal{A}} \| \boldsymbol{S}_A^n \|_1$, $C_{\bar{\boldsymbol{\theta}}}^{\max} = \max_{A \in \mathcal{A}, n \in [N]} \| \bar{\boldsymbol{\theta}}_A^n \|_2$.

From Eq.(54), Eq.(55), Eq.(57) and the definition of our sketched policy, letting $C_Y = \left( \kappa_{\text{all}}^{\max} \sqrt{\xi^{-2} - 1} \right) C_{\boldsymbol{S}} C_{\bar{\boldsymbol{\theta}}}^{\max}$, we obtain that

$$
\begin{aligned}
&\text{Reg}(\{A_{I_{n,b}}\}_{n \in [N], b \in [B]}) \\
&= \sum_{n \in [N], b \in [B]} \left[ \max_{A \in \mathcal{A}} \langle \boldsymbol{\theta}_A^*, \boldsymbol{s}_{n,b} \rangle - \left\langle \boldsymbol{\theta}_{A_{I_{n,b}}}^*, \boldsymbol{s}_{n,b} \right\rangle \right] \\
&\leq \sum_{n \in [N], b \in [B]} \left[ \max_{A \in \mathcal{A}} \left( \left\langle \tilde{\boldsymbol{\theta}}_A^n, \boldsymbol{s}_{n,b} \right\rangle + \alpha \sqrt{\boldsymbol{s}_{n,b}^{\mathsf{T}} (\boldsymbol{W}_A^n)^{-1} \boldsymbol{s}_{n,b}} \right) + Y_{n,b} - \left\langle \boldsymbol{\theta}_{A_{I_{n,b}}}^*, \boldsymbol{s}_{n,b} \right\rangle \right] \\
&= \sum_{n \in [N], b \in [B]} \left[ \left\langle \tilde{\boldsymbol{\theta}}_{A_{I_{n,b}}}^n, \boldsymbol{s}_{n,b} \right\rangle + \alpha \sqrt{\boldsymbol{s}_{n,b}^{\mathsf{T}} \left( \boldsymbol{W}_{A_{I_{n,b}}}^n \right)^{-1} \boldsymbol{s}_{n,b}} + Y_{n,b} - \left\langle \boldsymbol{\theta}_{A_{I_{n,b}}}^*, \boldsymbol{s}_{n,b} \right\rangle \right] \\
&\leq 2\alpha \sum_{n \in [N], b \in [B]} \sqrt{\boldsymbol{s}_{n,b}^{\mathsf{T}} \left( \boldsymbol{W}_{A_{I_{n,b}}}^n \right)^{-1} \boldsymbol{s}_{n,b}} + 2 \sum_{n \in [N], b \in [B]} Y_{n,b} \\
&\leq 2\alpha C_{\text{reg}} \sqrt{B} \sum_{n \in [N]} \sqrt{\sum_{b \in [B]} \boldsymbol{s}_{n,b}^{\mathsf{T}} \left( \boldsymbol{\Psi}_{A_{I_{n,b}}}^n \right)^{-1} \boldsymbol{s}_{n,b}} + 2N C_Y \sqrt{\rho_\lambda \epsilon d} \\
&= 2\alpha C_{\text{reg}} \sqrt{B} \sum_{n \in [N]} \sqrt{\sum_{b \in [B]} \left\langle \boldsymbol{s}_{n,b} \, \boldsymbol{s}_{n,b}^{\mathsf{T}}, \left( \boldsymbol{\Psi}_{A_{I_{n,b}}}^n \right)^{-1} \right\rangle} + 2N C_Y \sqrt{\rho_\lambda \epsilon d} \\
&= 2\alpha C_{\text{reg}} \sqrt{B} \sum_{n \in [N]} \sqrt{\sum_{A \in \mathcal{A}} \left\langle \boldsymbol{S}_A^{n\mathsf{T}} \boldsymbol{S}_A^n, (\boldsymbol{\Psi}_A^n)^{-1} \right\rangle} + 2N C_Y \sqrt{\rho_\lambda \epsilon d} \\
&= 2\alpha C_{\text{reg}} \sqrt{B} \sum_{n \in [N]} \sqrt{\sum_{A \in \mathcal{A}} \text{tr} \left( \boldsymbol{S}_A^{n\mathsf{T}} \boldsymbol{S}_A^n (\boldsymbol{\Psi}_A^n)^{-1} \right)} + O(N \sqrt{\rho_\lambda \epsilon d}), \\
&\leq 2\alpha C_{\text{reg}} \sqrt{BM} \sum_{n \in [N]} \sqrt{\max_{A \in \mathcal{A}} \left\{ \text{tr} \left( \boldsymbol{S}_A^{n\mathsf{T}} \boldsymbol{S}_A^n (\boldsymbol{\Psi}_A^n)^{-1} \right) \right\}} + O(N \sqrt{\rho_\lambda \epsilon d}).
\end{aligned}
\tag{58}
$$

When the structural assumption in Theorem 3 is not satisfied, from Eq.(41), we can obtain that the second term in Eq.(58) is also of order $O(\sqrt{\rho_\lambda \epsilon d})$, which does not influence the order of the final regret bound. Finally, combining Eq.(58) with lemma 3 in (Han et al., 2020) gives the final regret bound. □

## C  DETAILED EXPERIMENTAL SETTINGS AND MORE EXPERIMENTAL RESULTS

In this section, we provide more details and results in the experiments.

### C.1  DESCRIPTION OF DATASETS

Table 3 summarizes the description of datasets used in the experiments.

Next, we provide more details about the three datasets.

**Synthetic Data.** Inspired by the experiments in (Saito et al., 2020), the synthetic data generation procedure was formulated as follows, which simulates the streaming recommendation environment.

- Context $\boldsymbol{s}_i \in \mathbb{R}^d$: we drew elements of $\boldsymbol{s}_i$ independently from a Gaussian distribution $\mathcal{N}(0.1, 0.2^2)$, where $d = 40$;
- Click-Through-Rate (CTR): the CTRs for the 5 actions were respectively set as $\{10\%, 15\%, 25\%, 20\%, 30\%\}$;

Table 3: Description of datasets in the experiments ($T$: number of instances; $B$: batch size; $N$: number of episodes; $d$: dimensionality of context; $M$: number of actions; $C_B$ satisfying $B = C_B^2 N/d$)

| Dataset | $T$ | $B$ | $N$ | $d$ | $M$ | $C_B$ |
|---|---|---|---|---|---|---|
| synthetic data | 126,000 | 1,400 | 90 | 40 | 5 | 25.00 |
| Criteo-recent | 75,000 | 1,000 | 75 | 50 | 5 | 25.82 |
| Criteo-all | 1,276,000 | 4,000 | 319 | 50 | 15 | 25.04 |
| commercial product | 216,568 | 1,700 | 128 | 50 | 5 | 25.83 |

- The indicator variables of click events:

$$C_i = \begin{cases} 1 & \text{a click occurs in context } \boldsymbol{s}_i, \\ 0 & \text{otherwise.} \end{cases}$$

We sampled the click index set according to the uniform distribution.

- Conversion rate (CVR) in context $\boldsymbol{s}_i$: when $C_i = 1$,

$$\text{CVR}(\boldsymbol{s}_i) := \text{sigmoid}(\langle \boldsymbol{w}_c, \boldsymbol{s}_i \rangle), = \frac{1}{1 + \exp(-\langle \boldsymbol{w}_c, \boldsymbol{s}_i \rangle)},$$

where the coefficient vector $\boldsymbol{w}_c \in \mathbb{R}^d$ is sampled according to a Gaussian distribution as $\boldsymbol{w}_c \sim \mathcal{N}(\kappa_c \mathbf{1}_d, \sigma_c^2 \boldsymbol{I}_d)$, and we set different means and standard deviations for different action with $\kappa_c \in [0 : -0.2 : -0.8]$ and $\sigma_c \in [0.01 : +0.01 : 0.05]$;

**Criteo Data.** We used the publicly available Criteo dataset[5], consisting of Criteo's traffic on display ads over a period of two months (Chapelle, 2014), where each context consists of 8 integer features and 9 categorical features. Following the experiments in (Yoshikawa & Imai, 2018), the categorical features were represented as one-hot vectors and then concatenated to the integer features. We reduced the dimensionality of the feature vectors to 50 using principal component analysis (PCA). All of the algorithms were tested in a simulated online environment that was trained on users' logs in the Criteo dataset. Specifically, we chose several campaigns from the Criteo dataset, where each campaign represents a category of items and corresponds to an action. This online environment contains a prediction model for the CVR, which was well trained by applying DFM (Chapelle, 2014) using the true user feedbacks. This environment model was trained for each chosen campaign, whose AUCs are ranging from 70% to 90%, assuring that the online environment can provide nearly realistic feedbacks. To simulate the uncertainty of user behaviors, Gaussian noises with zero-mean were added to the model parameters. At each step, the online environment randomly selected a campaign and samples one context from this campaign, and revealed the context to the agent with a preset CTR. To generate a reasonable sequence of instances, the environment kept the order of timestamps of the contexts in each campaign. We tested our algorithms and the baselines with the following two online environments on the Criteo dataset: `Criteo-recent` contains 5 campaigns (75,000 instances) chosen from the recent campaigns, corresponding to 5 actions; `Criteo-all` contains 15 campaigns (1,276,000 instances) chosen from all the campaigns, corresponding to 15 actions.

**Data Collected from a Real Commercial App for Coupon Recommendation.** To verify the effectiveness and efficiency of our algorithms on real products, we conducted experiments on a real dataset collected from a commercial social app. We call this dataset `commercial product`, where the data were collected after the users gave consent, and did not contain any personally identifiable information or offensive content. Since this dataset from a commercial app is proprietary, we did not provide a URL. We will release this dataset after the publication of this paper. In this commercial app, after clicking a recommended coupon, a user may convert the coupon after some time, or just leave it there. The dataset was collected during a 1-month period with a subsampling, and consists of 216,568 instances from 5 categories of coupons, where each context is described by 86 numerical features and 16 categorical features. The timestamps of clicks and conversions were

---

[5]https://labs.criteo.com/2013/12/conversion-logs-dataset/

also recorded. Following the settings on the Criteo data, we also represented the categorical features as a one-hot vector, reduced the dimensionality of the feature vectors to 50 by PCA. The action space contains 5 actions, where each corresponding to one coupon category. Due to the limitation of real online experiments, in this experiment, we still trained DFM using the true user feedbacks as the online environment, where AUCs range from $75\%$ to $90\%$.

To simulate the real environment under partial-information feedback, The experiments were conducted in environments where the distribution of the initialization data is atypical. Specifically, in the experiments, we set different numbers of the initialization instances for each action. In the synthetic environment, we set the number of the initial instances as $140, 210, 350, 280, 420$ for the 5 actions, respectively. In `Criteo-recent`, we set the proportion of the initial instances as $0.1, 0.15, 0.25, 0.2, 0.3$ for the 5 actions, and set the number of the initial instances as $[100 : 23 : 423]$ for the 15 actions in `Criteo-all`.

## C.2 DETAILED SPECIFICATION OF HYPERPARAMETERS

In these experiments, the true reward is defined by $R_{i,A}^{\text{true}} = \lambda_c C_{i,A} + (1 - \lambda_c)V_{i,A}$ ($C_{i,A}$ and $V_{i,A}$ denote true binary variables of user click and conversion when executing action $A$ given context $s_i$), where $\lambda_c = 0.01$ on the synthetic data, Criteo Data, and commercial product data, respectively. As in most contextual bandit literature (Li et al., 2010; Chu et al., 2011), we set the regularization parameter $\lambda = 1$ in the Euclidean regularization. According to theoretical analysis in Remark 3, we set the batch size as $B = C_B^2 N/d$, set the constant $C_B \approx 25$ and the sketch size $c = 150$ on all the datasets ($B = 1400, 1000, 4000, 1700$ for `synthetic data`, `Criteo-recent`, `Criteo-all`, and `commercial product`). The regularization parameters $\omega$, $\alpha$ in our policy and that in the batch UCB policy were tuned in $[0.2 : +0.2 : 1.2]$. For the SJLT in SPUIR and its variants, sketch size was set as $c = 150$ and the number of block $D$ was selected in $\{1, 2, 4, 6\}$. Except for the rate-scheduled variants of our approaches, the imputation rate $\gamma$ was selected in $[0.1 : +0.2 : 0.9]$. Besides, the discount parameter $\eta$ was tuned in $[0.1 : +0.2 : 0.9]$. In the nonlinear variant of our approach SPUIR-Kernel, we selected the dimension of the random features $d_r$ in $\{50, 100, 200\}$ and the kernel width of Gaussian kernel in $\{2^{-(i+1)/2}, i = [-12 : 2 : 12]\}$.

**Rate-Scheduled Approach.** We equip PUIR and SPUIR with a rate-scheduled approach, called PUIR-RS and SPUIR-RS, respectively. We design a rate-scheduled approach following the theoretical results about the imputation rate $\gamma$. From Remark 1&2, we can obtain that a larger imputation rate $\gamma$ leads to a smaller variance while increasing the bias. From Remark 4, we conclude that the additional bias term includes a monotonic decreasing function w.r.t. number of episodes under mild conditions. Therefore, instead of using a fixed imputation rate, we can gradually increase $\gamma$ with the number of episodes, avoiding the large bias at the beginning of the reward imputation while achieving a small variance. Specifically, we set $\gamma = X\%$ for episodes from $(X - 10)\% \times N$ to $X\% \times N$, where $X \in [10, 100]$.

## C.3 MORE EXPERIMENTAL RESULTS

For better illustration, in Figure 3 of the manuscript, we omitted the curves of algorithms whose average rewards are $5\%$ lower than the highest reward. Now we provide the curves of all the algorithms in Figure 5.

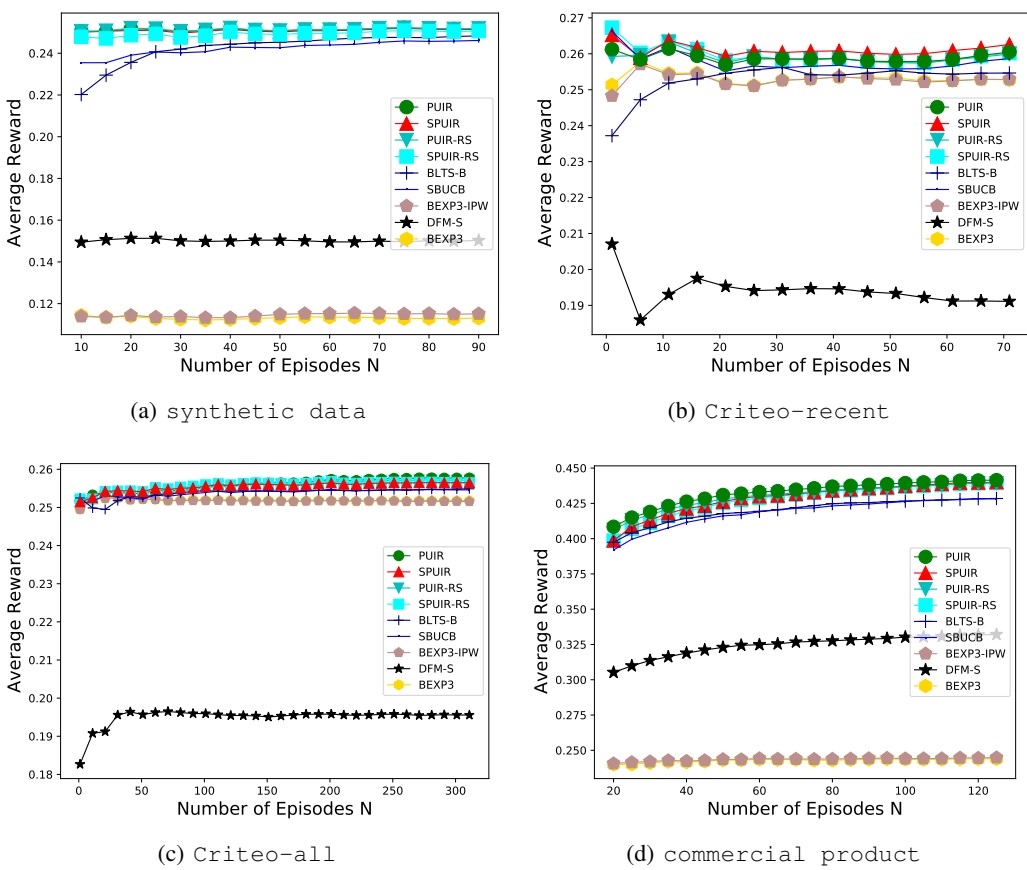

Figure 5: Average rewards of the compared algorithms, the proposed SPUIR and its variants on synthetic dataset, Criteo dataset, and the real commercial product data

