# OpenReview forum: "Partial Information as Full: Reward Imputation with Sketching in Bandits"
_ICLR.cc/2022/Conference — ICLR 2022 Submitted_

### Official Review · Reviewer_FYMj · 2021-10-24

**Correctness:** 2
**Technical Novelty And Significance:** 3
**Empirical Novelty And Significance:** 3
**Recommendation:** 5
**Confidence:** 4

**Main Review:**

PROS:
The theoretical work is complex and, besides the comment I make in the major comments section, accurate as far as I can tell. The paper has compared its approaches to a number of sensible benchmarks on a nice mix of simulated and real dataset and for several realistic extensions of the model. The problem studied is of genuine interest and the approach taken to derive a new algorithm is an interesting combination of ideas from the literature.

MAJOR COMMENTS ON CONS:
My main issues are that the explanations of the main algorithm are not sufficiently detailed to convince me that it is as effective as suggested. There is also an error in the theory which once corrected weakens the contributions. See more details below.

•	Construction of the imputation regularized ridge regression:
o	The paper is mostly accessible (with some omissions of detail as described in the more minor comments below) until this point where it stops being as user-friendly. Some more textual explanation of the terminology is needed here to help guide the reader as to the function and interpretation of matrices L, T, \Phi, \Psi, and b. This kind of discussion is provided prior to equation (1).

o	The jump from equation (1) to equation (2) is not obvious and could do with some further explanation.

o	Related to this, it’s not obvious where $\eta$ comes from – is it a function of $\gamma$ and $\lambda$? If not why does it not enter explicitly in to eq (1) somewhere? Are there infinitely many solutions and a particular choice of \eta parameterises a specific one?

o	At a more conceptual level, it’s hard to understand what this complex process of imputation achieves that is more than just reducing the importance of the regularization term and increasing the importance of the data-driven term. Since the imputed values for a batch are deterministic functions of the observed data in that batch, is the entire imputation process not equivalent to a very specific means of reducing $\lambda$?

•	Comments on the high-level idea of reward imputation and the exploration-exploitation trade-off:
o	Something that came to my mind upon reading the abstract but I couldn’t find directly addressed in the paper is the idea that reward imputation could feasibly push an algorithm towards exploitation and away from exploitation. I’m imagining a scenario where we draw a small number of samples for a particular action in an early round which are not particularly representative of its true value and lead to a biased estimate. Through deterministic imputation of the rewards that would have been observed on the other contexts, this bias will propagate further and would seem to only encourage the algorithm to favour this action less. If this happens to be a good action there would be a concern that this could lead to sticking in suboptimal actions and incurring more regret than algorithm which does not impute on the basis of biased estimates.

o	A particular concern in this setting would be that it leads to heavy tails on the regret distribution (as the sticking will only happen sometimes) and by reporting only means and standard deviations, this phenomenon is somewhat obscured.

•	THIS ISSUE HAS BEEN RESOLVED IN THE NEW VERSION: Error in the theory at eq (19)
o	I believe, unfortunately, that there is an error at eq (19) in the proof of Theorem 2. It has (nested within a square root) for $\eta \in (0,1)$ the equality \[
\sum_{i=0}^N \eta^{N-i} = \eta^N(1-\eta^{N+1})/(1-\eta)
\]

Which is not correct. It should be, with some simplification added for clarity,

\[
\sum_{i=0}^N \eta^{N-i} = 1+\sum_{i=1}^N \eta^N = 1+(1-\eta^N)\(1-\eta).
\]

This unfortunately cannot then be bounded as a decreasing function of N and, I believe, leads to a constant order additional bias, perhaps bounded as $(C_\theta \gamma (1+1/(1-\eta)))$ in the final statement of the Theorem.

If I am not mistaken, and there is not some other technique that can be used to remove this additional bias – from a conceptual viewpoint this seems unlikely as it makes sense for a reduction in variance to be necessarily traded off against an increase in bias – this necessitates some modifications to the discussion and reframing of the contributions. The exponentially decreasing bias is sold as a major contribution (indeed arguably oversold, as it is only the additional bias which would exponentially vanish, not the entire bias), and therefore the loss of this does affect the message of the paper somewhat.


MINOR COMMENTS ON CONS:
•	The related work section may be improved and made more useful by indicating how these papers link to your work (principles of making a batched decision, optimism vs other methods, similarities or inspiration in the theoretical analyses).

•	The sketching approach is quite an important concept to the paper, but one that is not commonly used in bandits and therefore potentially unfamiliar to prospective readers of the paper. It would be nice to make it clearer what it involves (conceptually) when it’s first mentioned in the introduction.

•	I don’t think $M$ is defined? Further to this, it may be useful to give a sense of the likely values of $M, B, d$ early in the paper? For instance, I found myself worrying about situations where B is close to M and some actions are not used at all in some early rounds which seems to be less relevant when you get later in the paper and realise the intended setting is M<<B?

•	The definition of a policy $\pi$ could be more mathematically precise – the text seems to suggest that $\pi_n$ is a single sampling distribution on $\mathcal{A}$ used for all elements of the n^th batch. But it is not clear how precisely it maps from the contexts and history to actions, and whether it can vary across the batch depending on, say, the earlier selections in the batch.

•	The distribution of the rewards is not made clear in section 2, only the functional form of their expectation – is this deliberate?

•	The experiment in Figure 1 isn’t fully reproducible, as the experimental parameters (reward distributions, context distribution, and batch size) aren’t specified. This also makes it harder to interpret what level of feedback the algorithm could expect without any additional information. Finally, it’s not clear what modifications are made to the Han et al. algorithm to adapt to the action-specific reward parameters, or whether this is all conducted in the setting of a shared reward parameter.

•	On page 4 you say “for $\forall A_j$” which reads as saying ‘for’ twice.

•	From Section 4 the amount of whitespace seems to have been reduced between paragraphs? This makes things quite dense and hard to read.

•	At the start of the proof of theorem 2: *triangle inequality

•	I don’t think you need the $s \neq 0$ condition in Thm 2 since $\Gamma$ is positive semi-definite not positive-definite?


**Summary Of The Paper:**

This paper addresses batched contextual bandits, with a fixed set of actions, and separate unknown parameters for each action. The authors propose an approach where the unobserved rewards (i.e. the rewards that would have been obtained if actions that hadn’t been selected for a given context) are imputed, and these imputed values are incorporated in to the regularization of the parameter estimates in a Lin-UCB-like algorithm. For reasons of computational efficiency, the authors also design a process to approximate this regularized estimator via a ‘sketching’ technique. For the approaches with and without sketching, the authors derive a $O(\sqrt{dMT})$ regret bound, where M is the number of actions, and d is the dimensionality of the parameter vectors, which broadly matches what is expected in the non-batched setting. They argue that the proposed algorithms have uniformly lower variance in the instantaneous regret than approaches without imputation and any increase in bias decreases exponentially quickly. Versions with time-adaptive parameters, and for non-linear functions are proposed without a full theoretical treatment, and all the proposed algorithms are shown to perform well in an empirical study.

**Summary Of The Review:**

There is a substantial amount of work which has gone into this paper, and the authors have combined several different ideas – contextual bandits, batched bandits, sketching approximations for efficiency – together in a mostly careful manner. This is non-trivial work. My concerns are that there several areas where the explanations feel limited and not user friendly, where the interesting features of the method are not fully explained, and where there are slip-ups in the theory and reproducibility. The latter two of these points mean that I’m not fully convinced of the effectiveness and utility of the method.

I also question whether ICLR is the right venue for this work. It seems that it has been difficult to adequately explain the method, its theoretical guarantees and its place within the literature within 9 pages, and a more accessible discussion may be achievable in a journal paper.

---

> ### Author Response · Authors · 2021-11-14
> **Response to Reviewer FYMj (1/2)**
>
> # To all reviewers
>
> Thanks for your comments and suggestions. We have revised the manuscript according to the detailed comments. The modifications are as follows:
> 1. We have rewritten the process of the proposed reward imputation process to make it clearer to the readers.
> 2. We have refined regret analyses in Theorem 2, proved a new lemma for analyzing the additional
> bias of reward imputation (Lemma 1 in Appendix B.1), and provided a thorough analysis for controlling the additional bias (Remark 2 and Remark 4).
> 3. We have revised the symbols and their description according to the comments.
>
> We would appreciate if the reviewers kindly let us know of any leftover concerns in the very limited time remaining. We would be happy to do our utmost to address them.
>
> # Response to Reviewer FYMj
>
> **Construction of the imputation regularized ridge regression**
>
> *1. The specification of parameter*
>
> In the updated manuscript, we have added the discussion about the block matrix prior to equation (1), as well as the solution (2). Besides, $\eta$ is a hyper-parameter for online updating the context and reward matrix, thus we do not contain it in the regression equation. We have added a thorough analysis for controlling the additional bias (Remark 2 and Remark 4), and the conclusion is that the additional bias is monotonic decreasing w.r.t. number of episodes provided that the mild condition $\sqrt{\eta} = \Theta(d^{-1})$ holds, where $d$ is the dimensionality of inputs.
>
> *2. The use of imputation process*
>
> The proposed imputation process is not used for reducing the importance of the regularization term. The proposed approach of imputation is introduced for estimating the unobserved feedbacks that are not contained in each batch of data. From the theoretical and experimental analyses in Theorem 1 and Figure 1, we can observe that when the unobserved feedbacks are correctly received for policy updating, agent can achieve higher rewards. This motivates us to propose a computationally efficient and theoretically sound imputation approach for estimating the unobserved feedbacks in an online manner.
>
> **High-level idea**
>
> *1. exploration-exploitation trade-off*
>
> Exploration–exploitation dilemma is the key challenge in online learning under bandit settings. In the full-information setting, UCB policy does not need to explore and achieves a lower variance part in the regret (Theorem 1). Along this line, our reward computation approach is proposed to approximate the setting of full-information feedback, which also brings a lower variance part and a controllable additional bias part in the regret.
>
> *2. Adversarial data*
>
> In this paper, we focus on the stochastic bandit setting, and a deterministic policy is sufficient to achieve an optimal regret. The goal of bandit policy is to approximate the optimal policy in hindsight. While the environment is adversarial, e.g., a small number of samples for a particular action in an early round which are not particularly representative, the optimal policy can only achieve a low cumulative reward that is the upper bound of any policy.
>
> *3. Regret distribution*
>
> The variance and error in regret are terminologies in contextual bandit literature, which do not correspond to the statistics of regret distribution. Regret analysis can be seen as a kind of convergence rate analysis in online setting, which focuses on the order of the regret bound.
>
> **Proof of Theorem 2**
>
> Thanks for pointing out the mistake in the inequality. In the updated manuscript, we have revised this mistake and provided a new analysis of the additional bias. Specifically, we have given a refined bound of the bias (inequality (26)), proved a new lemma for analyzing the additional bias of reward imputation (Lemma 1 in Appendix B.1), and provided a thorough analysis for controlling the additional bias (Remark 2 and Remark 4). Our analyses conclude that, the additional bias term is controllable, and the additional bias contains a monotonic decreasing function w.r.t. number of episodes provided that a mild condition holds (the corresponding analyses can be found in Appendix B.1). Besides, we have also modified the corresponding statements about the theoretical results and contributions in the manuscript.

---

> > ### Author Response · Authors · 2021-11-14
> > **Response to Reviewer FYMj (2/2)**
> >
> > **Number of actions $M$**
> >
> > We do not need to restrict the number of actions $M \ll B$. In Theorem 4, the condition is $M = O(\mathrm{poly}(d))$, where a large $M$ does not influence the order of the regret against the optimal policy in hindsight.
> >
> > **Distribution of the rewards**
> >
> > We focus on the stochastic bandit setting, where only the expectation of the reward is specified (typically defined as a linear function).
> >
> > **Setting of experiments in Figure 1**
> >
> > As stated in the caption of Figure 1, we conducted this experiment following the same synthetic environment in Section 6, where the experimental parameters (reward distributions, context distribution, and batch size) are specified in Appendix C.2.
> > Due to the space limitation, as stated in Algorithm 3 in Appendix A.2, we had given the description of what modifications are made to the Han et al. algorithm to adapt to the setting of action-specific reward parameters (i.e., our CBB setting).
> >
> > **Related work and others**
> >
> > We have added a discussion about the sketching technology in related work. Besides, we have revised other symbols and their description according to the comments.

---

> > > ### Comment · Reviewer_FYMj · 2021-11-20
> > > **Reply to Author Response**
> > >
> > > Thank you for taking the time to respond to my comments.
> > >
> > > I feel there are, unfortunately, still outstanding issues subsequent to these replies and the revision.
> > >
> > > The point I was making about the imputation process reducing the importance of the regularisation term was not one around intention or purpose but around what using this imputation approach practically achieves. As there is more data (deterministically added via imputation), for any given value of the regularisation parameter, this regularisation term should be less important in the optimisation than otherwise. While I do not go so far as to say that this renders the algorithm in anyway ineffective - I believe the theoretical guarantees are now sound - I do think it is a point of note, and that in the context of balancing exploration and exploitation, it is something that there should be a discussion around. The mathematical descriptions are quite dense, and it's hard for the user to get a firm grasp on why this imputation technique will not cause overfitting or the algorithm to commit to bad actions. It's a fair point that there is less need for exploration when full feedback is observed, but the imputed data is only recycling the limited feedback observed.
> > >
> > > I'm pleased the authors have been able to identify a version of the theory which corrects for the error in Theorem 2's proof. But I feel that in the short time that has been given to remedy this, a clear discussion of the balance between bias and variance has not yet been achieved. Like the use of imputation, the rate scheduling of $\gamma$ seems to now be a really key, interesting feature of the algorithm and analysis and I don't feel that it is afforded the insightful, clear discussion it needs at the moment. I think the paper needs more rewriting to make really clear to the reader what the challenges in this design are, and why the solutions used are the right ones which avoid potential pitfalls.
> > >
> > > In terms of the high-level idea, I'm not talking about adversarial data, rather the event with non-zero probability that the initialisation data is atypical for the distribution: the kind of thing that UCB typically guards against and that an exploration-exploitation trade-off is required for. Issues with respect to this are not necessarily represented in a theoretical treatment of the expected regret, but can be observed in experimental results, and the empirical distribution of regret over multiple trials, which is what I am referring to when I speak of heavy tails.
> > >
> > > I think there is an interesting idea underpinning this paper, and it provides an opportunity for a rich discussion of the exploration-exploitation trade-off in contextual bandits. I still feel however that the present draft does not push things far enough to reach this yet.

---

> > > > ### Author Response · Authors · 2021-11-22
> > > > **Response to Reviewer FYMj**
> > > >
> > > > # Response to Reviewer FYMj
> > > >
> > > > Thanks for your response and constructive suggestions. We have addressed all comments and suggestions in the below section and made appropriate changes to the manuscript in this revision.
> > > >
> > > > **Exploration and exploitation trade-off**
> > > >
> > > > Following your suggestion, we have added a discussion about the relationship to the exploration and exploitation trade-off in Remark 5. But we respectfully disagree with your comment ‘this paper needs a rich discussion of the exploration-exploitation trade-off’. The reason is that, exploration and exploitation trade-off is an **intuitive explanation** among many perspectives for studying online learning under limited feedback. In this paper, our work is built around the **regret analysis**, which is a **theoretical perspective** for this topic. We analyzed the difference of regrets between the full-information case and the partial-information case, indicating the motivation of the reward imputation, designed new algorithms based on the conditions of sublinear regret. Our goal is to find a refined trade-off between the bias and variance in regret.
> > > >
> > > > **Discussion of the rate scheduling**
> > > >
> > > > In ‘Detailed Specification of Hyperparameters’ in the new version, we have added a discussion about the setting of the rate-scheduled approach. Besides, in Remark 2, Remark 3, and Remark 4, we think that we have clarified the challenges in balancing the bias and variance, and discussed how we meet these challenges in algorithm design.
> > > >
> > > > **Initialisation data**
> > > >
> > > > The experiments were actually conducted in an environment where the distribution of the initialisation data is atypical (shown in ‘create_data’ in code) that corresponds to reviewer’s concern. Specifically, in the experiments, we set different numbers of the initialisation instances for each action, where the numbers of the initial instances are set as 140, 210, 350, 280, 420 for the 5 actions in the synthetic environment, respectively. Besides, our experiments on the real-world datasets were performed under a similar initialization about the atypical distribution. The experiment results show that, in the presence of the unbalanced feedback at the beginning of the sequential-decision, the proposed reward imputation approach can also help in improving the average reward. We have added the above discussion in the ‘description of datasets’ of the updated version.

---

> > > > > ### Comment · Reviewer_FYMj · 2021-11-26
> > > > > **Raising to 5**
> > > > >
> > > > > I'm in agreement with the authors that the paper has improved since its initial version, and agree with the authors with regards to their comments about exploration-exploitation versus regret analysis. Ultimately, I don't feel that these points are made as clearly in the paper as in their discussions, and that the paper needs further revision, not in terms of the substance of the contributions, but in the manner that they are presented.
> > > > >
> > > > > To recognise the improvements, and in light of the discussion I will increase my score to a 5, as I no longer feel it is a clear reject.

---

### Official Review · Reviewer_cFbQ · 2021-11-03

**Correctness:** 3
**Technical Novelty And Significance:** 2
**Empirical Novelty And Significance:** 3
**Recommendation:** 5
**Confidence:** 4

**Main Review:**


Positive points:
* the paper is well written and generally easy to follow;
* the results are clearly communicated; both means and standard deviations are provided (e.g in Tables);
* in my biased opinion, the problem proposed  and the chosen approach are both interesting;


Major issues:
* details regarding the imputation procedure itself were scarce and hard to parse;
* the effect of different imputation methods could be rather large, this goes unaddressed in the current paper;
* comparisons are performed with methods where the parameters are shared across arms/actions. While this assumption is restrictive application wise, it is important to understand the benefits, if any, of imputation in those cases as well;
* there is little intuition regarding the support/rank of the context matrix, arm specific parameters and their interaction, and how this can impact  whether imputing contexts is helpful or not;
* simple Thompson Sampling seems to be performing comparatively well;

Minor issues:
* the number of iterations used to compute the means and standard deviations should be made explicit (10? 100?);
* for the top performing algorithms, it would be interesting to see the standard deviation bands throughout the plots;
* claiming that most literature ignores potential rewards is slightly misleading: Bayesian methods aim to model distributions over  rewards given contexts, hence there is an implicit
* occasional spelling

Suggestions for improvement
* assess the impact of imputation when different imputation methods are considered, be explicit about how the imputation is achieved
* assess the performance of the algorithms in settings where the arm/action parameters are the same across arm/actions
* consider synthetic experiments where the distributions of contexts and action parameters are made explicit, and where the rewards are modeled as explicit functionals of context and actions. I believe [1] could be a great basis for such experiments.
* address minor issues;


[1] Lloyd, James, et al. "Random function priors for exchangeable arrays with applications to graphs and relational data." Advances in Neural Information Processing Systems 25 (2012): 998-1006.

**Summary Of The Paper:**

The paper considers the behavior of a series of algorithms for learning in a batched bandit setting,
where parameters are arm/action specific, rather than shared across arms/actions.

These algorithms are based on UCB like techniques with the addition of a regularization term that aims
to use information from imputed, unobserved rewards. Such techniques are present in the linear regression
and dimensionality reduction literature where one aims to use either prior information or
data distribution information to aid prediction. However, to my knowledge, these formalisms are quite
new to online settings like bandits, thus making the presented approach appealing.

**Summary Of The Review:**

Recommendation: weak reject.
The paper addresses an interesting question, and is generally well written, with a promising algorithmic direction. However,
I believe the central imputation idea gets lost in the many variants considered. I believe having sketching variants
is important and I liked that facet of the paper. However, I believe that there isn’t sufficient clarity on how the actual imputation is accomplished, and what is the impact of such imputation. Further, the paper does not provide intuition on what modeling assumptions and statistical data features are behind the performance improvements when imputation is used.

---

> ### Author Response · Authors · 2021-11-14
> **Response to Reviewer cFbQ**
>
> # To all reviewers
>
> Thanks for your comments and suggestions. We have revised the manuscript according to the detailed comments. The modifications are as follows:
> 1. We have rewritten the process of the proposed reward imputation process to make it clearer to the readers.
> 2. We have refined regret analyses in Theorem 2, proved a new lemma for analyzing the additional
> bias of reward imputation (Lemma 1 in Appendix B.1), and provided a thorough analysis for controlling the additional bias (Remark 2 and Remark 4).
> 3. We have revised the symbols and their description according to the comments.
>
> We would appreciate if the reviewers kindly let us know of any leftover concerns in the very limited time remaining. We would be happy to do our utmost to address them.
>
> # Response to Reviewer cFbQ:
>
> Thanks for your comments and constructive suggestions.
>
> **Imputation procedure**
>
> We have rewritten the part of the imputation procedure, and provided a new analysis of the additional bias for our reward imputation approach. Specifically, we have given a refined bound of additional bias of reward imputation (inequality (26)), proved a new lemma for analyzing this additional bias (Lemma 1 in Appendix B.1), and provided a thorough analysis for controlling the additional bias (Remark 2 and Remark 4).
>
> **Effect of different imputation methods**
>
> Imputation methods are presented in the linear regression and dimensionality reduction literature, where the fitting process is conducted in an offline manner. Appling the imputation method to online learning in bandit settings is complicated and needs to be careful, since bandit setting needs an efficiently incremental updating process and sublinear regret guarantees. We do not think every imputation method can be applied to the setting we considered, and developed a reward imputation approach specially tailored for this bandit setting.
>
> **Comparisons with the case where the parameters are shared**
>
> The proposed reward imputation approach can be directly applied to the case where the parameters are shared. Specifically, we only need to maintain one reward parameter vector in each episode, and apply it to every imputed reward. We focus on the more complex setting where the parameters are action specific, since we find that real product dataset usually shares the context for each action and has action-specific reward mechanisms.
>
> **Help of imputing contexts**
>
> The key step for constructing the imputed contexts is to multiply the received contexts by the discount parameter $\sqrt{\eta}$ in each episode. In the updated manuscript, we have added the discussion the impact of the parameter $\sqrt{\eta}$ in Remark 2 and Remark 4, and the conclusion is that the additional bias in the regret is monotonic decreasing w.r.t. number of episodes provided that the mild condition $\sqrt{\eta} = \Theta(d^{-1})$ holds, where $d$ is the dimensionality of inputs.
>
> **The performance of BLTS-B**
>
> BLTS-B is not a simple Thompson Sampling algorithm. BLTS-B is a revised linear contextual version of the Thompson Sampling algorithm, which uses inverse propensity score for modifying the observed feedbacks.
>
> **Experimental settings**
>
> As stated in the experimental settings in Section 6, we repeated each experiment 20 times for computing the means and standard deviations.
>
> **Synthetic experiments**
>
> Thanks for your suggestions about the synthetic experiments. As described in Appendix C.1, in our synthetic experiment, the distributions of contexts, action parameters, and rewards were made explicit according to some known functions.

---

### Official Review · Reviewer_yEx1 · 2021-11-08

**Correctness:** 4
**Technical Novelty And Significance:** 3
**Empirical Novelty And Significance:** 3
**Recommendation:** 6
**Confidence:** 4

**Main Review:**

Pros:
The paper is well-written and considers an important problem of batched linear bandits in the literature.

(A) The authors introduce the idea of imputation to further speedup the learning process. Regret bounds showing that the lower variance of the imputation approach over the vanilla version further strengthens the paper.
(B) Sketching is then introduced to further speed up the computation of the regression problem. The introduced sketching error is further tuned by appropriately setting the sketch size to provide a sublinear regret of O(\sqrt{MDT}).
(C) Experimental results are pretty convincing on the provided datasets of both improved regret performance as well as in terms of sketching time. Better regret performance is achieved by SPUIR compared to other approaches but roughly less than half the  time of PUIR.

Questions:

(1) Why are the Theta's for each of the actions in the episodes computed afresh? Could we reuse the previously computed theta's to regularize the solution in the current batch and get rid of the imputation which is expensive?
(2) This uses the classic ``sketch and solve'' paradigm. Can we utilize the ``iterate and sketch'' to further speed up the approach?[a]

[a] Oblivious Sketching-based Central Path Method for Linear Programming. ICML 2021

**Summary Of The Paper:**

The paper considers the contextual batched bandit setting and introduces the idea of imputation utilizing the non-executed actions in each batch. This provides better regret properties than without and also further speedup is provided by considering the sketch version.
Theoretical results in terms of sketching performance are also provided such as going down from O(Bd^2) to O(cd^2) for sketch size c as well as the regret bounds of the sketched approach SPUIR.
Experimental results are shown on a couple of datasets to showcase the improved performance over state-of-the-art batched bandit algorithms such as BEXP3, BLTS-B.

**Summary Of The Review:**

Overall, the paper solves an important problem in the literature of contextual batched bandits. The proposed approach of imputation of unobserved actions is sound and both theoretical results in terms of regrets bounds and superior practical performance on real-world datasets are shown in the paper.
Still unsure about the imputation approach taken and if we can further improve the sketching approach considered in the paper.

---

> ### Author Response · Authors · 2021-11-14
> **Response to Reviewer yEx1**
>
> # To all reviewers
>
> Thanks for your comments and suggestions. We have revised the manuscript according to the detailed comments. The modifications are as follows:
> 1. We have rewritten the process of the proposed reward imputation process to make it clearer to the readers.
> 2. We have refined regret analyses in Theorem 2, proved a new lemma for analyzing the additional
> bias of reward imputation (Lemma 1 in Appendix B.1), and provided a thorough analysis for controlling the additional bias (Remark 2 and Remark 4).
> 3. We have revised the symbols and their description according to the comments.
>
> We would appreciate if the reviewers kindly let us know of any leftover concerns in the very limited time remaining. We would be happy to do our utmost to address them.
>
> # Response to Reviewer yEx1:
>
> Thanks for your appreciation of the gist of the paper.
>
> **Computation of $\theta$**
>
> When we want to reuse the previously computed $theta$, a stochastic optimization method must be introduced for solving the proposed imputation regularized ridge regression. This stochastic optimization framework can avoid the high computational complexity of imputation, but it may lead to a large optimization error since the error will be accumulated and propagated as the number of episodes increases. Thus, in this paper, we use a closed-form solution for solving this regression problem, avoiding the large optimization error. How to control the optimization error in bandits is an interesting topic and we will leave it for further works.
>
> **Use of other sketching approaches**
>
> In the proposed reward imputation, we choose SJLT as the sketch matrix due to its good incremental and approximate properties. We can also use other sketching approaches in our sketching approach and we will leave this topic for further works.

---

### Decision · Program_Chairs · 2022-01-20

**Decision:**

Reject

**Comment:**

In this paper the authors consider a contextual batched bandit setting where they rely on  imputationin order to estimated the non-executed actions in each batch. Even though the idea is quite ineteretsing, and can lead to new methods, there is still a lof of issues raised by the reviwers. In particular, part of the proof was incorrect (and the authors tried to fix it) but given the short time, the reviwers felt that this part should be rewritten and scrutanized further. Also, there are many suggestions by reviewers that the authors need to apply in order to make this work publishable.